# Whole-genome sequencing reveals sex determination and liver high-fat storage mechanisms of yellowstripe goby (*Mugilogobius chulae*)

Lei Cai [1,3✉], Guocheng Liu[2,3], Yuanzheng Wei[1,3], Yabing Zhu [2,3], Jianjun Li[1], Zongyu Miao[1], Meili Chen[1], Zhen Yue [2], Lujun Yu[1], Zhensheng Dong[2], Huixin Ye[1], Wenjing Sun[2] & Ren Huang [1✉]

As a promising novel marine fish model for future research on marine ecotoxicology as well as an animal model of human disease, the genome information of yellowstripe goby (*Mugilogobius chulae*) remains unknown. Here we report the first annotated chromosome-level reference genome assembly for yellowstripe goby. A 20.67-cM sex determination region was discovered on chromosome 5 and seven potential sex-determining genes were identified. Based on combined genome and transcriptome data, we identified three key lipid metabolic pathways for high-fat accumulation in the liver of yellowstripe goby. The changes in the expression patterns of *MGLL* and *CPT1* at different development stage of the liver, and the expansion of the *ABCA1* gene, innate immune gene *TLR23*, and *TRIM* family genes may help in balancing high-fat storage in hepatocytes and steatohepatitis. These results may provide insights into understanding the molecular mechanisms of sex determination and high-fat storage in the liver of marine fishes.

---

[1] Guangdong Provincial Key Laboratory of Laboratory Animals, Guangdong Laboratory Animals Monitoring Institute, Guangzhou, China. [2] BGI Genomics, BGI-Shenzhen, Shenzhen, China. [3]These authors contributed equally: Lei Cai, Guocheng Liu, Yuanzheng Wei, Yabing Zhu. ✉email: cailei17@163.com; 1649405216@qq.com

Compared with most mammals, sex determination mechanisms in fish have exhibited a high degree of plasticity and complexity[1,2], which may be related to genetic or environmental factors or both. So far, several sex determination systems have been identified in fishes, including male-heterogametic gonochorism (XY)[3], female-heterogametic gonochorism (ZW)[4], hermaphroditism[5], and environmental dependency[6]. However, the reasons for the evolution of so many diverse sex determination mechanisms and the key factor for transition of different sex determination mechanisms remain unknown[7]. Especially in gobiid fish, one of the largest fish families, comprising more than 2000 species, little information about the sex determination mechanism has been discovered, and no sex determination genes or sex determination regions have been identified to date[8–10]. Considering that gobiid fishes are one of the most important taxa in the marine ecosystem, understanding their sex determination mechanism is of great significance in revealing the adaptive strategies and providing invaluable insights on the evolution of sex determination in teleosts.

Lipids and their constituent fatty acids are major organic constituents in various organisms from worms (Caenorhabditis elegans) to humans, and the ability to store fats is conserved[11,12]. Interestingly, starting with the primitive teleosts (jawless vertebrates such as lampreys), the lipid-storing cells have evolved into a tissue that has distinct functions underneath the skin[13], while the type and sites of fat storage is species-specific in fish and depend on the nutritional state, life-stage, and the physiological state[14,15]. Most fishes, such as zebrafish (Danio rerio)[14] and cavefish (Astyanax mexicanus)[16], show deposition of lipids, mainly triglycerides (TAG), in mesentery and viscera. However, in majority of the gobiids, the lipids are only stored in the liver[17–20]. This phenomenon is also exhibited in some other marine fishes, such as pufferfish (Takifugu rubripes) and cultured flounder (Paralichthys olivaceus)[21]. The special lipid deposition organ in these fish might be an evolutionary adaptation to cope with the typical living environments, such as rapid energy mobilization, migration, or benthic adaptation. This is similar to the situation observed among the Inuits, who display genetic and physiological adaptations to a diet rich in polyunsaturated fatty acids (PUFAs)[22] and stickleback lineages (Gasterosteus aculeatus species complex), which have evolved different copy numbers of lipid metabolism-related genes, such as docosahexaenoic acid biosynthesis-related genes, to achieve transitions between marine and freshwater environments[23]. In the majority of gobiids, large quantities of neutral fat are stored in the liver. In most gobiid fishes, fat represents more than 70% of the liver wet weight, and 90% of the total lipids are triglycerides[18]. In humans, the liver fat content is less than 5% of the wet weight[24]. When this percentage is exceeded, non-alcoholic fatty liver disease (NAFLD) can develop, and more than 20% of patients with NAFLD develop non-alcoholic steatohepatitis (NASH)[25] along with inflammation and varying degrees of fibrosis[26]. However, gobiids can maintain a high level of fat storage in the liver lifelong without developing steatohepatitis, suggesting the existence of a specialized mechanism for maintaining a balance between high-fat storage and inflammation in the liver in these fishes. Hence, elucidating the high-fat storage mechanism of gobiids is of great value for understanding the evolution of lipid storage in teleosts, and the pathogenesis of human NASH as well as for promoting aquaculture health in the mariculture industry.

Gobiids (Teleostei, Gobiidae), commonly known as gobies, are a diverse and fascinating group with worldwide distribution[27], and is one of the most diverse families of vertebrates on earth[28]. Therefore, gobies represent a potential excellent model for adaptation studies. Unfortunately, neither the genome sequence and phylogenetic relationships of many groups of gobies nor the laboratory breeding and rearing methods are resolved. Only a few gobiid genomes have been sequenced, such as those of round goby (Neogobius melanostomus)[8], mudskippers (Boleophthalmus pectinirostris, Periophthalmodon schlosseri, Periophthalmus magnuspinnatus, Scartelaos histophorus)[10], and sand goby (Pomatoschistus minutus)[29]. Currently, only a few small fish species [e.g., zebrafish[30], Japanese medaka (Oryzias latipes)[31], and platyfish (Xiphophorus maculatus)[32]] have been widely used in the laboratory; however, the large majority of these inhabit freshwater environments. Laboratory models of marine species are limited to the three-spined stickleback species (Gasterosteus aculeatus)[33], pufferfish (Takifugu rubripes)[34], and the Atlantic silverside (Menidia menidia)[35]. Considering the specific needs of laboratory animals, such as convenient large-scale indoor cultivation and controlled year-round spawning, it is necessary to develop a representative marine fish to supplement the marine model fish species. Yellowstripe goby (Mugilogobius chulae) is a representative fish of the Gobiidae family that is widely distributed along the western Pacific coast[36]. This species has a moderate body size (adult body length, 3–5 cm), short sexual maturity period, strong reproductive capacity, short spawning interval, annual reproduction, easy indoor rearing, and easy genetic manipulation[37]. In addition, a Chinese national quality-control standard, including genetic, microorganism, parasite, nutrition, and environment quality control, has been established for yellowstripe goby (draft national standard no. 20091329-T-469)[38]. Hence, the yellowstripe goby is a quality-controlled laboratory fish and a promising novel marine fish model for future research on genetic evolution and marine ecotoxicology as well as an animal model of human disease.

In this study, a 7th generation inbred line of yellowstripe goby was subjected to whole-genome sequencing using Illumina HiSeq and PacBio RSII sequencing platforms. A high-density genetic linkage map was constructed based on single-nucleotide polymorphism (SNP) markers, using restriction site-associated DNA (RAD) sequencing. The constructed genetic map was used to assemble the genome at the chromosomal level. A sex determination region was identified by quantitative trait locus analysis. Further, based on combined genome and transcriptome data, we aimed to identify the potential mechanism underlying neutral fat storage and lipid homeostasis in the liver of yellowstripe goby.

## Results

**Genome assembly and annotation.** Using a 7th generation inbred line (female), we generated 134.9 giga base pairs (Gb) of clean reads (135× coverage) by Illumina short-read sequencing and 28.7 Gb of clean reads (29X coverage) by PacBio long-read sequencing (Supplementary Table 1), which were corrected and hybrid-assembled into 7098 contigs and 1776 scaffolds, respectively. The yellowstripe goby reference genome was 1.002 Gb, with a scaffold N50 of 1.57 Mb, and a contig N50 of 261 kb (Table 1).

The GC distribution of the genome was relatively concentrated and unbiased (Supplementary Fig. 1). The GC content of yellowstripe goby (39%) was similar to that of the great blue-spotted mudskipper (Boleophthalmus pectinirostris), zebrafish, and Japanese medaka (Supplementary Fig. 2). BUSCO evaluation revealed that the assembled genome contained 87% of the known fish orthologous genes (Supplementary Table 2). When mapping the assembled transcriptome[37] to the assembly, 99% of the sequences were mappable (Supplementary Table 3), indicating a high-quality assembly. Using 65.29 Gb of sequencing data from the HiSeq platform for 17-mer analysis, the heterozygosity of the yellowstripe goby genome was calculated to be 1.2% (Supplementary Fig. 3).

**Table 1 Statistical analysis of the genome-assembly results.**

| Genome assembly | | N50 | Max length(bp) | Total length(bp) | Number | |
|---|---|---|---|---|---|---|
| | Contigs | 260,505 | 2,174,452 | 988,921,936 | 7098 | |
| | Scaffolds | 1,569,707 | 9,301,748 | 1,002,319,200 | 1776 | |
| Protein-coding genes | | Total number | | annotated | unannotated | |
| | | 20,531 | | 19,729 | | 802 |
| Non-coding RNAs | | Copy | Average length (bp) | | Percentage of genome (%) | |
| | miRNA | 367 | 84.11 | | 0.0031 | |
| | tRNA | 1273 | 74.46 | | 0.0095 | |
| | rRNA | 5328 | 504.75 | | 0.0306 | |
| | snRNA | 226 | 394.22 | | 0.0028 | |
| TEs | | Trf | Repeatmasker | Proteinmask | De novo | Total |
| | Number | 49,843,024 | 66,868,036 | 26,540,334 | 438,150,562 | 467,641,374 |

We identified 20,531 protein-coding genes (Supplementary Tables 4 and 5) in yellowstripe goby, which was similar in number to the 20,798 protein-coding genes in great blue-spotted mudskipper[10], but lower than those in zebrafish (26,260)[30] and round goby (*Neogobius melanostomus*) (38,773)[8]. The lengths of mRNAs, coding sequences, exons, and introns in the yellowstripe goby genome were consistent with those in mudskipper, zebrafish, and Japanese medaka (Supplementary Fig. 4). The yellowstripe goby genome had an overall repeat content of 42.56%, which is similar to that in mudskipper[10] and round goby[8], but lower than that in zebrafish[30] (52.2%) and Atlantic salmon[39] (58%). Long interspersed nuclear elements (LINEs) (15.88%) and DNA transposons (15.61%) were the most enriched repeat elements, whereas short interspersed nuclear elements (SINEs) were the least prevalent (2.26%) (Supplementary Table 6 and Supplementary Fig. 5).

**Construction of a high-density SNP-based genetic linkage map and assisted genome assembly.** RAD sequencing was carried out for the two parents and 225 F1 progenies, and 381 Gb of clean data, with an average sequencing depth of approximately 30× were obtained. Reads were aligned to the reference genome to identify SNP sites. We identified 627,394 and 666,860 SNPs in the female and male parent, respectively. After filtration, we constructed a high-density genetic linkage map of yellowstripe goby (Supplementary Fig. 6) based on 9534 SNP markers, representing the first genetic linkage map for the family Gobiidae. The total map length was 3098.2 centimorgans (cM), and the average genetic distance between markers was 0.32 cM, which is higher than that in most fishes at present. Using the genetic linkage map, we anchored the assembly to 22 chromosomes (Fig. 1a), containing 1065 scaffolds and 922 Mb (92%) of the total length of the assembled sequences, representing the first genome assembled at chromosomal scale in the family Gobiidae.

**Comparative genome analysis.** Nineteen representative species were selected for phylogenetic analysis; 17,347 gene families were identified, and 572 single-copy orthologous genes were selected for phylogenetic tree construction and divergence time estimations (Fig. 1b). The phylogenetic trees revealed that yellowstripe goby and mudskipper diverged 77 million years (Myr) ago and yellowstripe goby diverged from other teleosts ~120 Myr ago, which was later than the time of mudskipper divergence (140 Myr)[10].

We identified 12,088 orthologous genes among yellowstripe goby, zebrafish, and humans, and 6609 genes were common among the three species (Supplementary Data 1, Fig. 1c). Yellowstripe goby shared more orthologous genes (7354) with zebrafish than did grass carp (*Ctenopharyngodon idella*) (7227)[40].

The spectrum of synonymous substitutions (Ks) among yellowstripe goby, mudskipper, and *Oncorhynchus mykiss* showed peaks at 0.5 for gobiids (yellowstripe goby versus mudskipper, yellow curve in Fig. 1d), which were close to those for trout (*O. mykiss* versus *O. mykiss*, pink curve in Fig. 1d), that has undergone four whole-genome duplication events (WGD) (Fig. 1d). However, the peak with yellow line continued into another peak with green line (yellowstripe goby versus yellowstripe goby) (Fig. 1d). Thus, yellowstripe goby has just undergone three WGD events. We compared the reference genome of yellowstripe goby with Japanese medaka; most chromosomes (20/22) of the yellowstripe goby showed a one-to-one relationship with the medaka chromosomes, and only two chromosomes of yellowstripe goby showed a one-to-two correspondence with the medaka chromosomes (Fig. 1a), indicating that the relationship between their genomes was high.

**Sex determination mechanism.** We compared RAD sequencing data for males and females to identify sex determination regions in yellowstripe goby. Notably, we detected a strong signal (log of the odds score = 12.5) with a 20.67-cM-broad peak on chromosome 5 (Fig. 2), representing 49.2 Mb physical size. This is the first sex determination region to be discovered in the family Gobiidae. In this region, 58 SNPs and 102 genes were identified.

We selected the 25 most associated SNPs (Supplementary Table 7) to genotype 200 random wild fish samples ($n = 100$ males and $n = 100$ females) (Supplementary Methods). The locus numbered S247-888402 was homozygous (GG) in all female fish and had three genotypes (AA, GG, AG) in male fish (Supplementary Table 8). Sequence analysis showed that locus S247-888402 was located in the second intron of the *GALNT10*-like (polypeptide N-acetylgalactosaminyltransferase 10-like) gene. *GALNT10*-like expression was higher in the ovary than in the testis (Supplementary Table 9, Supplementary Fig. 7a). Functional annotation of the 102 genes revealed two more genes that may be related to sex determination, namely *MSL3* (male-specific lethal 3 homolog) and *H2AFY* (core histone macro-H2A.1) (Supplementary Table 10).

Illumina sequencing was used to identify genes differentially expressed between the testis (SA) and ovary (OV) of yellowstripe goby (Supplementary Methods). We found two male-determining genes and one female-determining gene, namely, doublesex and mab-3-related transcription factor 1 (*DMRT1*), gonadal soma-derived factor (*Gsdf*), and forkhead box L2 (*FOXL2*), respectively. *DMRT1* was expressed only in the testis, while *Gsdf*, which is a downstream-regulated gene of *DMRT1*, was highly expressed in the testis (Supplementary Table 9). *FOXL2* is a critical gene for female determination and was highly expressed in the ovary but was also expressed at low levels in the testis (Supplementary Table 9). In addition, a *FOXL2* homolog (*FOXL3*; forkhead box L3), which may

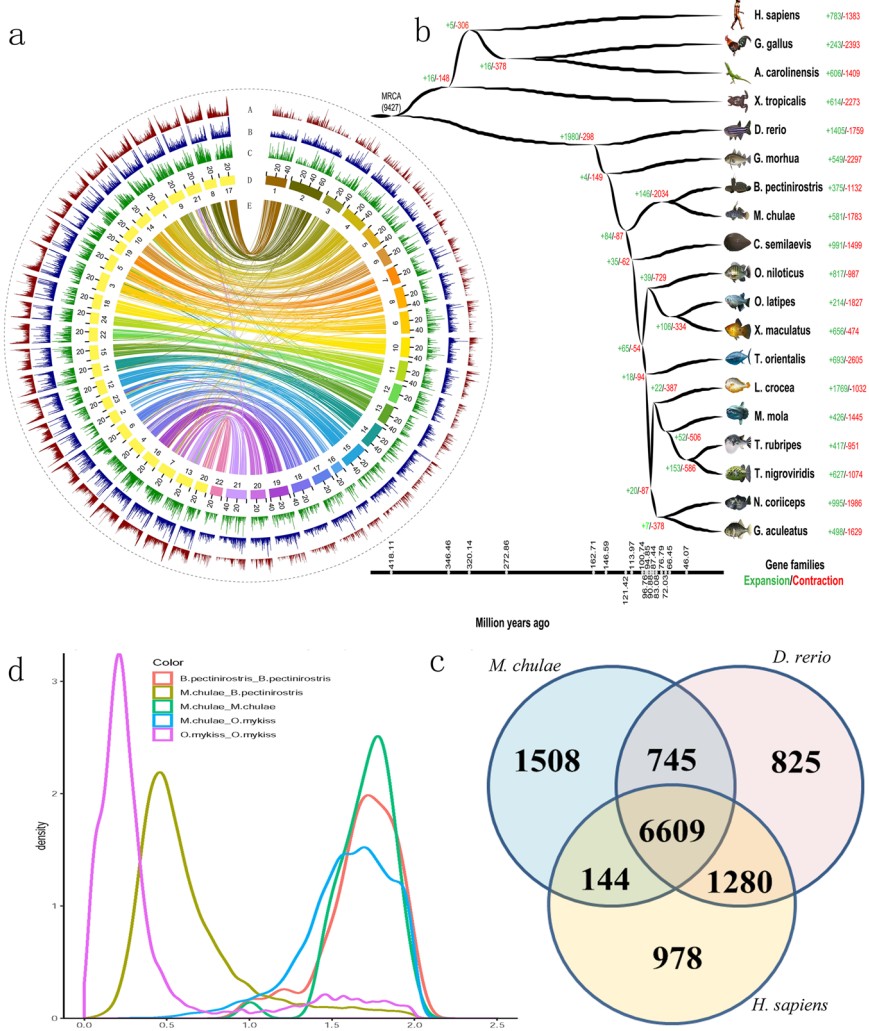

**Fig. 1 Yellowstripe goby genome features. a** Landscape of the 22 assembled yellowstripe goby chromosomes. From the outer to the inner: GC_content, ssr density, gene density, chromosomes, maps of the 22 yellowstripe goby chromosomes and of the 24 Japanese medaka chromosomes based on the positions of 11,756 orthologous pairs demonstrate highly conserved synteny for the 2 species. **b** The divergence-time tree of single-copy genes. **c** Gene families of *M. chulae*, *D. rerio*, and *H. sapiens*. **d** The third genome duplications in the yellowstripe goby genome was identified by Ks analyses; the pink curve represents the fourth genome duplication event of *O. mykiss*, green and orange curve represents the third genome duplication event of yellowstripe goby and mudskipper, yellow and blue curve represents the interspecific differentiation.

be related to testis and ovary development, was expressed only in the testis. The expression data are shown in Supplementary Table 9.

The expression of all these genes was validated by quantitative real-time PCR (RT-qPCR). Concordance between RNA-seq and RT-qPCR results was observed for all seven genes (Supplementary Fig. 7, Supplementary Methods).

**Global upregulation of lipid synthetic pathway genes might underlie liver high-fat storage**. Histological observation of the liver of yellowstripe goby showed that high-fat deposition in the liver might be a normal physiological phenomenon, according to histologic features in different developmental stages (Supplementary Fig. 8) and a 28-day starvation assay (Supplementary Fig. 9). The lipid component represented up to 77% of the liver wet weight (as indicated by mass spectrometry), which exceeds that in other species such as zebrafish, medaka, and humans by a large degree (Supplementary Table 11). Triglyceride was the main liver lipid, accounting for 92.59% of the total lipid dry weight (Supplementary Fig. 10; Supplementary Methods). Glucose

tolerance tests showed that glucose is cleared quickly from the blood in yellowstripe goby, as impaired glucose tolerance was not observed (Supplementary Fig. 11; Supplementary Methods). In addition, we did not observe inflammatory gene expression in normal hepatic tissues. Thus, the liver is a natural energy-storage organ in yellowstripe goby.

Liver fat rapidly accumulated between 10 and 60 days of age, and intrahepatic lipid droplets filled the hepatocytes after 3 months of age. Transcriptome analysis of livers from 2-month-old (G2M) and 3-month-old (G3M) fish revealed that lipid synthetic genes in the G2M group were globally upregulated compared with that in the G3M group (because of the low correlation of G3M-2 when compared with G3M-1 or G3M-3, the data for G3M-2 were discarded; Supplementary Fig. 12, Supplementary Tables 12 and 13, Supplementary Methods). These genes are mainly involved in the synthesis of TAG and cholesterol (CHOL), the two most important components in hepatocyte lipid droplets.

TAG is synthesized in hepatocytes via two major pathways, namely glycerol-3-phosphate (G3P) → lysophosphatidic acid

(LPA) → phosphatidic acid (PA) → diacylglycerol (DAG) → TAG and monoacylglycerol (MAG) → DAG → TAG. Our analysis revealed that some key genes involved in these two synthetic pathways, including glycerol-3-phosphate acyltransferase 3 (*GPAT3*), lipid phosphate phosphohydrolase 1 (*PAP1*), and diacylglycerol O-acyltransferase 2 (*DGAT2*) were upregulated in the livers from the G2M group compared with those from the

G3M group (Fig. 3, Supplementary Table 13). Concordance between RNA-seq and RT-qPCR results was observed (Supplementary Fig. 13, Supplementary Methods).

Another major component in lipid droplets is CHOL ester, which is mainly synthesized in hepatocytes or transported to hepatocytes from peripheral tissues. The main synthetic pathway in the liver is acetyl-CoA → mevalonate → squalene → CHOL,

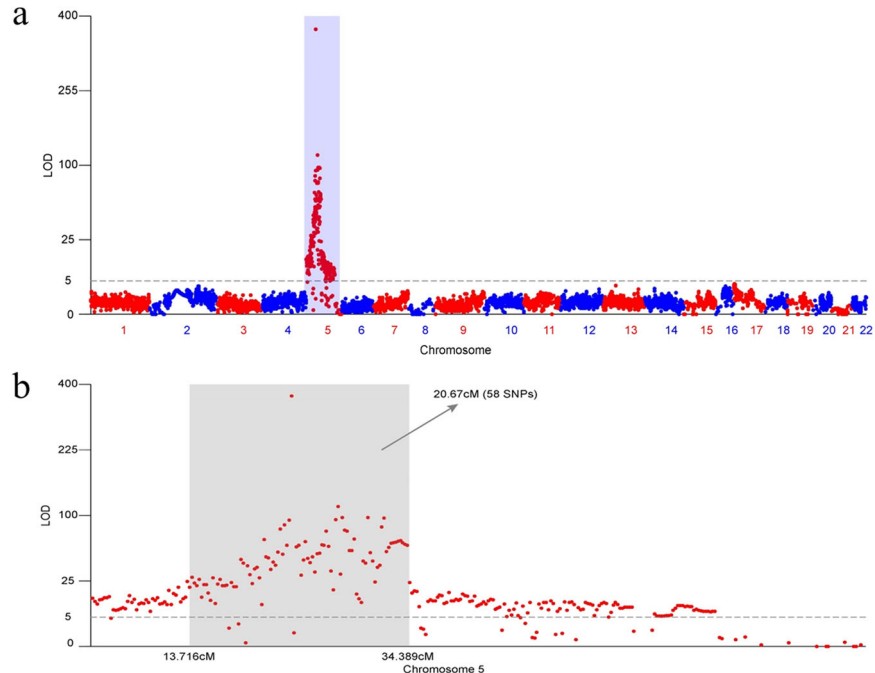

**Fig. 2 QTL mapping of sex linkage in yellowstripe goby. a** Genome-wide LOD (Likelihood ratio statistic) score for tests of genotype difference between sexes, arranged by chromosome. The gender-related signal is occurred only at chromosome 5. **b** A 20.67-cM-broad peak (sex-association QTL search) on chromosome 5.

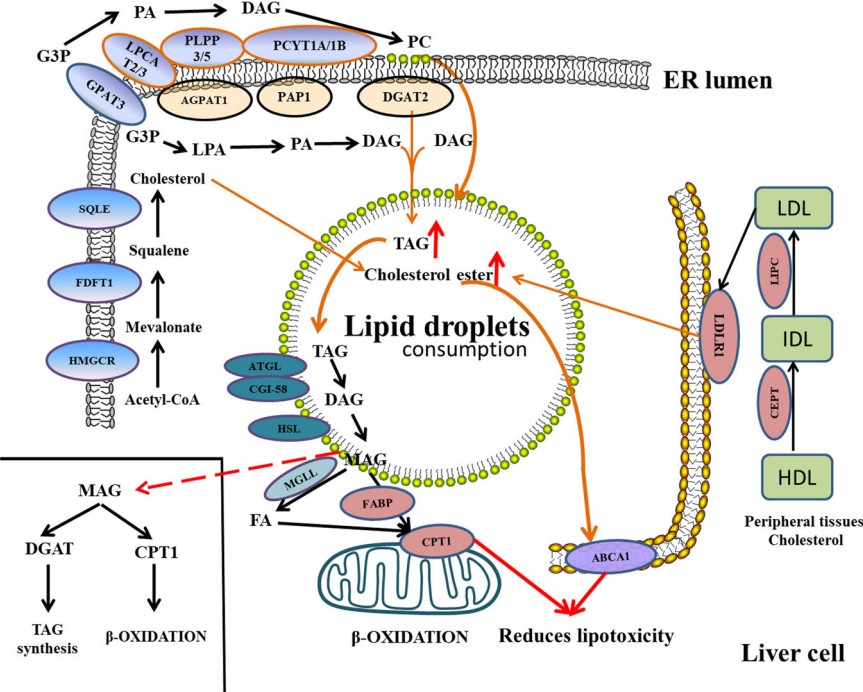

**Fig. 3 Formation and maintenance mechanisms of lipid droplets in the liver of yellowstripe goby.** Genes include *GPAT3, PAP, DGAT2* in TAG synthesis; *FDFT1, SQLE, LIPC, LDLR1* in CHOL synthesis; *LPCAT3* in phospholipid synthesis; *MGLL, CPT1,* and *ABCA1* in maintaining the balance between liver high-fat storage and steatohepatitis.

and the associated key genes *FDFT1* and *SQLE* were upregulated in G2M livers compared with those in the G3M livers (Fig. 3, Supplementary Table 13). High-density lipoprotein (HDL) is necessary for reverse cholesterol transport (RCT) from the peripheral to hepatic tissues. We found that genes associated with HDL transport, including those encoding hepatic lipase (*LIPC*), and low-density lipoprotein receptor adapter protein 1 (*LDLR1*) were all upregulated in G2M livers (Fig. 3, Supplementary Table 13). The increase in HDL RCT function in the G2M group might promote CHOL deposition in lipid droplets. Concordance between RNA-seq and RT-qPCR results was observed (Supplementary Fig. 13, Supplementary Methods).

**Differential expression of monoglyceride lipase (*MGLL*) gene might promote lipid accumulation in the early stage of the liver.** TAG decomposition in lipid droplets is mainly driven by patatin-like phospholipase domain-containing 2 (*ATGL*) and abhydrolase domain-containing 5 (*CGI-58*) on lipid droplet membranes to generate DAG, which is then degraded by the lipid droplet membrane protein, hormone-sensitive lipase (*HSL*), to produce MAG. MAG is finally degraded by intracellular MGLL to free fatty acids (FFAs) and glycerol. Excessive intracellular FFAs are essential for activating Kupffer cells, which induce cellular inflammation and NASH. Surprisingly, *MGLL* gene expression was low in the livers from the G2M group, whereas it was highly expressed in the livers from the G3M group. The low *MGLL* expression in the G2M group might contribute to the promotion of lipid accumulation in the early stage of the liver (Fig. 3).

Low expression of *MGLL* in G2M group hepatocytes may lead to drastic accumulation of the decomposition target, MAG. We found that MAG in hepatocytes could be returned to lipid droplets via MAG → TAG pathway. *DGAT2*, a key gene in the MAG → TAG pathway, was highly expressed in the G2M livers. In the G2M livers, MAG mostly was re-metabolized into TAG through *DGAT2* for storage. Moreover, the results of RNA-seq analysis could be verified by RT-qPCR (Supplementary Fig. 13, Supplementary Methods).

**Increased phospholipid synthesis contributes to maintaining lipid droplet membrane homeostasis.** Phospholipids play a key role in maintaining lipid droplet membrane homeostasis. Lecithin (PC) and cephalin (PE) are the main components of the mono-layer phospholipid membrane of lipid droplets. We found that the key genes required for PC and PE synthesis, including those encoding *GPAT3* and lysophosphatidylcholine acyltransferase 3 (*LPCAT3*) were upregulated in G2M compared to those in G3M (Fig. 3, Supplementary Table 13). Furthermore, concordance between RNA-seq and RT-qPCR results was observed (Supplementary Fig. 13, Supplementary Methods).

**Carnitine palmitoyltransferase 1 (*CPT1*) reduces lipotoxicity by promoting free fatty acid consumption in hepatocytes.** We found that *CPT1* expression was much higher in the G3M group than in the G2M group (Supplementary Table 13, Supplementary Fig. 13). After the TAG storage in hepatocytes reached saturation in the G3M group, MAG mainly underwent β-oxidation through mitochondrial *CPT1* for energy supply (Fig. 3). *CPT1* might reduce lipotoxicity by promoting FFA consumption in hepatocytes. Concordance was observed between RNA-seq and RT-qPCR results (Supplementary Fig. 13).

**ABCA1 expansion reduces lipotoxicity by increasing RCT function in hepatocytes.** Sequence analysis revealed that *ABCA1* of yellowstripe goby has the largest copy number among known species (Supplementary Table 14), with four copies distributed over four chromosomes (Fig. 4a). All *ABCA1* genes had two nucleotide-binding domains and two characteristic extracellular domains. Analysis of the tissue distribution of *ABCA1* revealed that *ABCA1a-1* and *ABCA1c* are the most highly expressed genes in the yellowstripe goby liver (Fig. 4b). Phylogenetic analysis revealed that *ABCA1c* was the ancestral copy (Fig. 4c), and most fishes had lost it during evolution, as had humans and zebrafish. *ABCA1* is a key RCT gene in hepatocytes. Expansion of yellowstripe goby *ABCA1* and the high *ABCA1c* expression in the liver, in particular, could reduce the lipotoxicity of CHOL to hepatocytes by increasing RCT in hepatocytes, thus maintaining the balance between liver high-fat storage and steatohepatitis.

**A robust innate immune system helps reduce the amplification effect of the external environment on steatohepatitis.** We discovered an expansion of *TLR23* (15 copies) and tripartite motif containing (*TRIM*) family members (234 members) in yellowstripe goby (Supplementary Tables 15 and 16). All *TLR23* copies had a Toll/IL-1 receiver (TIR) domain, and one of them (*TLR23h*) had two TIR domains, representing the first fish TLR gene with these characteristics (Fig. 5). A phylogenetic tree was constructed for *TLR23* genes, using the neighbor-joining method, based on the amino acid sequences of yellowstripe goby and mudskipper (Fig. 6). Two distinct clades, representing two subfamilies were clearly distinguished. The tripartite motif containing (*TRIM*) family is another important gene family involved in innate immunity. The *TRIM* family encodes E3 ubiquitin ligases, which are involved in several important biological processes, especially, antiviral responses. We identified 234 *TRIM* genes in yellowstripe goby, which is the largest number among known species (Supplementary Table 16), with *TRIM14*, *TRIM16*, *TRIM21*, *TRIM25*, *TRIM35*, and *TRIM39* being the most abundant (Fig. 7). Among them, *TRIM21* had the largest copy number (58) (Supplementary Table 17). Expansion of *TLR23* and the *TRIM* family might be beneficial in the context of high-fat storage.

## Discussion

We generated a whole-genome sequence of yellowstripe goby, using Illumina short-read and PacBio long-read sequencing. This represents the first annotated chromosome-level reference genome assembly for yellowstripe goby, providing important basic information for future research on the genetic evolution, sex determination, diseases, and marine ecotoxicology of yellowstripe goby as a potential marine model fish. Further, we constructed a high-density SNP genetic linkage map for yellowstripe goby, the first in the family Gobiidae (>2000 species). The average marker distance was 0.32 cm, which is lower than that for most non-model and non-aquaculture species, including *Cyprinus carpio haematopterus* (0.57 cm)[41], *Nibea albiflora* (0.47 cm)[42], *Larmichthys crocea* (0.36 cm)[43], and *Pseudobagrus ussuriensis* (0.36 cm)[44]. Using the constructed genetic map, the yellowstripe goby genome was assembled at the chromosome level, and >92% of the assembled sequences were anchored. This is the first genome assembled at the chromosomal level in the family Gobiidae[8–10].

We discovered that chromosome 5 of yellowstripe goby carries a 20.67-cm sex determination region that contains three genes that might be related to sex determination, namely, *MSL3*, *H2AFY*, and *GALNT10*-like genes. *MSL3* contributes to overexpression of genes on the X chromosome of male *Drosophila*[45]. Research on the structure of human *MSL3* has shown that it functions similar to the *Drosophila MSL3*[46], by binding to lysine 20 on the N-terminal tail of histone H4 to regulate the male-specific lethal complex on the X chromosome. *H2AFY* helps maintain X-chromosome inactivation[47]. Thus, *MSL3* and *H2AFY* may be involved in yellowstripe goby sex determination. *GALNT10*-like participates in

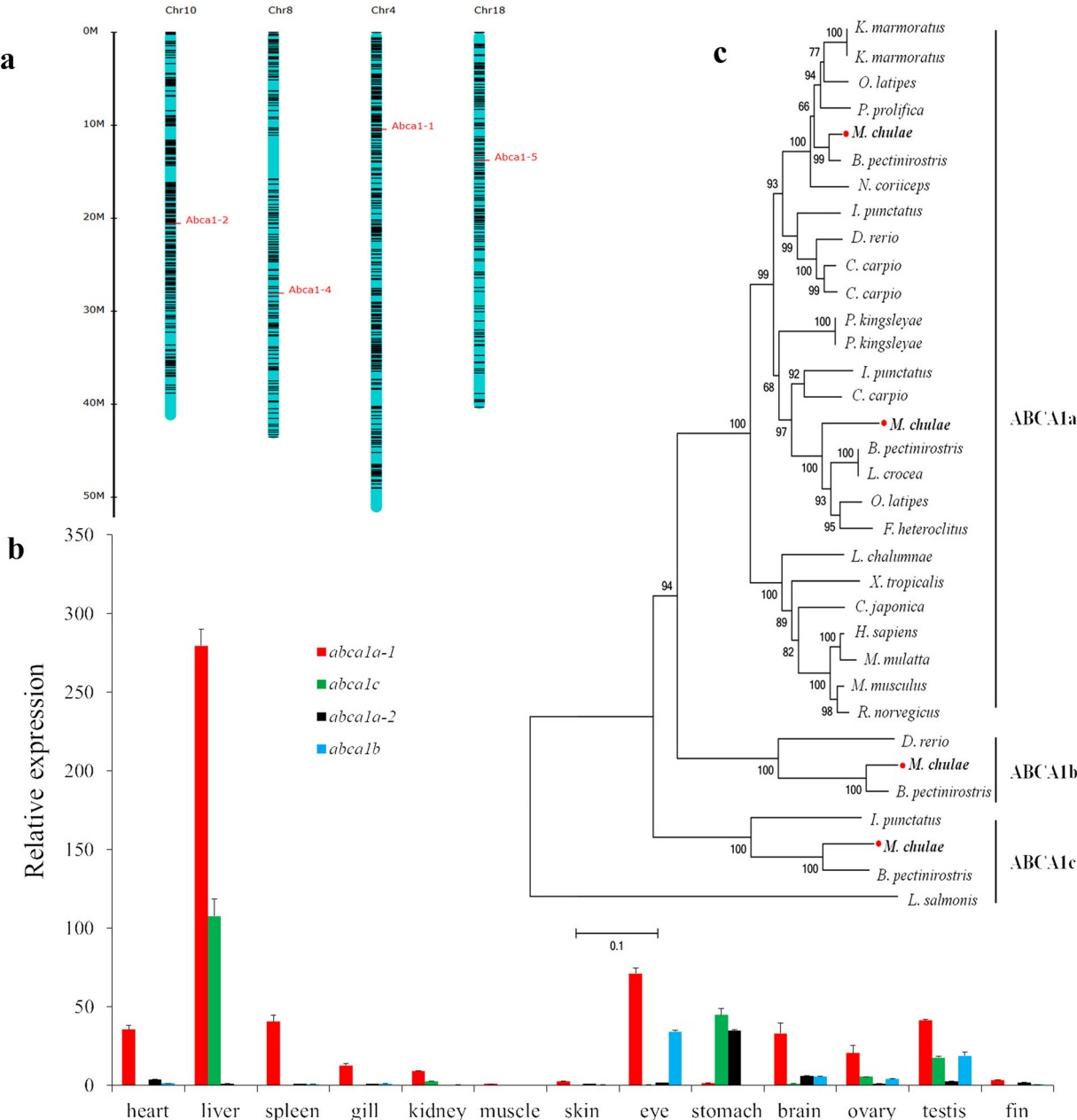

**Fig. 4 Structure and expression analysis of *ABCA1* in yellowstripe goby. a** Chromosomal location of *ABCA1* genes. **b** The relative expression of *ABCA1* genes in 13 tissues (*n* = 3 biologically independent tissue samples). **c** Phylogenetic tree of *ABCA1* between yellowstripe goby and other representative species.

the synthesis of mucin-type oligosaccharides[48]. We found a SNP that was located in the second intron of this gene to be homozygous (GG) in all female fish. Quantitative real-time PCR validation of *GALNT10-like* in different tissues of female and male individuals also revealed that it was highly expressed only in the ovary. Most fish eggs are surrounded by a glycoproteinaceous structure[49,50] called an egg envelope, which provides the embryo with physical protection. *GALNT10-like* might participate in the formation of the egg envelope and filament adhesion apparatus of yellowstripe goby. Sequence analysis showed that locus S247-888402 was located in the second intron of the *GALNT10-like* gene. The relationship between *GALNT10-like* gene and its SNP at S247-888402 with respect to sex determination might be worthy of further investigation. In addition, testis-specific *DMRT1* expression was detected based on the transcriptome and RT-qPCR data.

*DMRT1* is a master male sex-determining gene in various species, including chicken[51], *Xenopus laevis*[52], and *O. latipes*[53]. It induces gonadal differentiation by regulating *GSDF*, which was highly expressed in the testis. Thus, *DMRT1* and *GSDF* might be potential sex-determining genes in yellowstripe goby. In addition, we found that the *FOXL2* was highly expressed specifically in the yellowstripe goby ovary. *FOXL2* is a key gene in ovarian differentiation and development[54,55] and may participate in female sex determination in yellowstripe goby.

As a small bottom-layer fish inhabiting estuaries and intertidal zones, yellowstripe goby often faces sudden changes in salinity, temperature, water depth, light, and other factors. To facilitate rapid energy mobilization, yellowstripe goby has evolved a biological phenotype wherein the liver stores energy to cope with complex and changing living environments. Based on combined

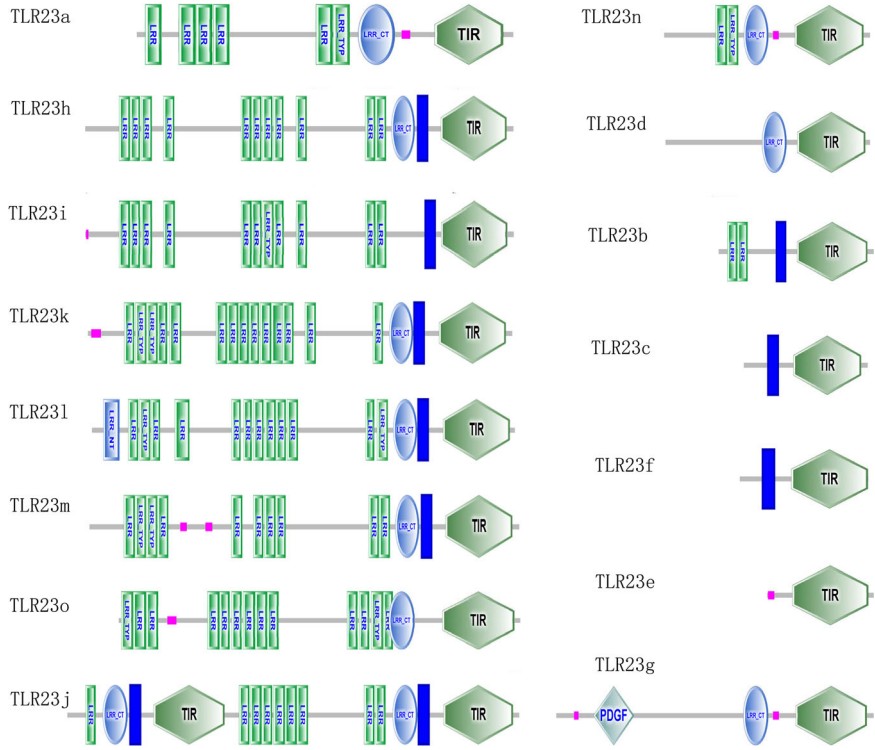

**Fig. 5 Protein domain structures of *TRL23* in yellowstripe goby.** LRR leucine-rich repeat, LRR-TYP leucine-rich repeat typical subfamily, TIR Toll/IL-1 receptor, LRR-NT leucine-rich repeat N-(nitrogen) terminal, LRR-CT leucine-rich repeat C-(carboxyl) terminal, PDGF platelet-derived growth factor.

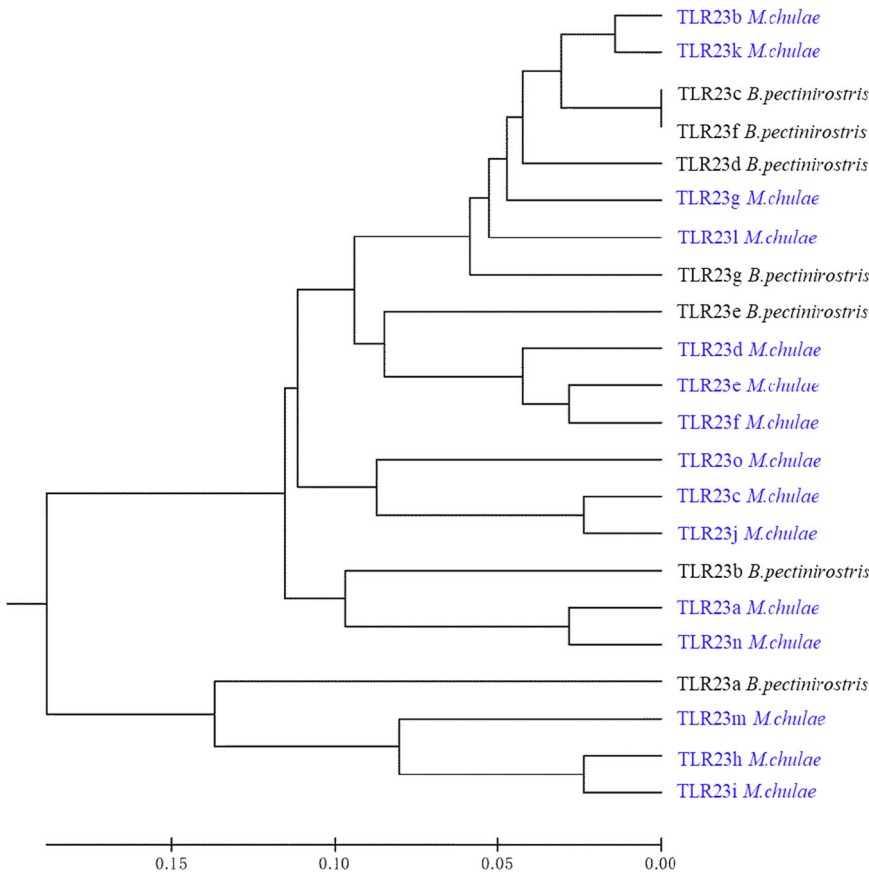

**Fig. 6 Phylogenetic tree of *TRL23* between yellowstripe goby and *B. pectinirostris*.** *TLR23* genes of yellowstripe goby are highlighted in blue. Phylogeny of *TLR23* family between yellowstripe goby and *B. pectinirostris* showing the expansion of *TLR23* in yellowstripe goby.

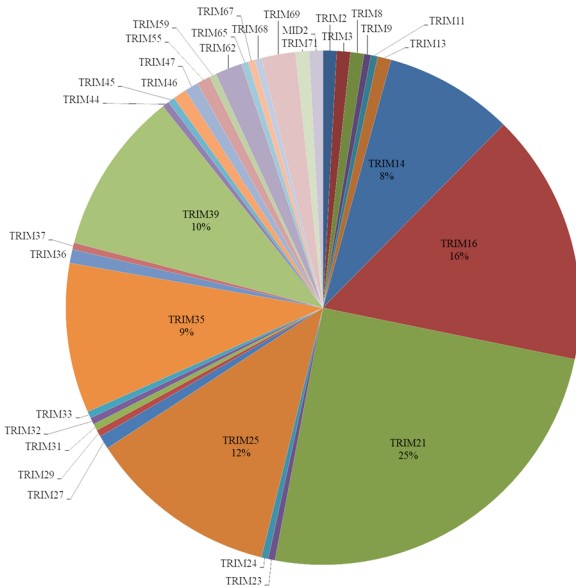

**Fig. 7 Distribution of gene numbers in the *TRIM* gene family in yellowstripe goby.** A total of 34 *TRIM* gene families (234 members) were identified in yellowstripe goby, which is the largest number among known species (Supplementary Table 16), with *TRIM14*, *TRIM16*, *TRIM21*, *TRIM25*, *TRIM35*, and *TRIM39* being the most abundant.

genome and transcriptome data, we identified three key metabolic pathways for lipid accumulation in the liver of yellowstripe goby, namely TAG synthesis, CHOL synthesis, and PC synthesis. TAG and CHOL are the two major components in the lipid droplets of hepatocytes, and PC is a major structural component of bio-membranes, including lipid droplet membranes. Key genes involved in TAG, CHOL, and PC synthesis were globally upregulated in the G2M livers; especially, the key genes *GPAT3*, *PAP1*, *DGAT2*, *FDFT1*, *SQLE*, *LIPC*, *LDLR1*, and *LPCAT3* were all expressed at higher levels in the G2M phase than in the G3M phase (Supplementary Fig. 13). Increased TAG and CHOL synthesis can promote lipid droplet accumulation, while increased PC synthesis is crucial for regeneration and maintaining the stability of lipid droplets[56,57]. High hepatic expression of genes associated with TAG, CHOL, and PC synthesis and transportation possibly promotes lipid droplet formation in the liver in yellowstripe goby.

In yellowstripe goby, the liver starts to accumulate TAG after hatching and maintains a high-fat accumulation state throughout life without an accompanying inflammatory reaction (Supplementary Fig. 8 and Fig. 9). Yellowstripe goby faces great challenges in maintaining normal physiological function and energy-storage balance in the liver. Genome and transcriptome analysis revealed that the yellowstripe goby might maintain the balance between lipid accumulation and intracellular inflammatory response by modulating the differential expression of *MGLL* and *CPT1*. Usually, TAG is decomposed into MAG by *ATGL*, *CGI-58*, and *HSL* on the surface of lipid droplets[58,59] (Fig. 3), and MAG is further decomposed into FFAs by *MGLL*[60]. Surprisingly, *MGLL* gene expression was low in the livers from the G2M group. The low *MGLL* expression in the G2M group might contribute to the promotion of lipid accumulation in the early stage of the liver (Fig. 3). In contrast, *MGLL* was highly expressed in the livers from the G3M group and might produce vast quantities of FFAs in the hepatocytes. FFAs are derived from lipids stored in lipid droplets and can produce severe lipotoxicity in cells[61]. We found that *CPT1* expression was much higher in the G3M group than in the G2M group (Supplementary Table 13, Supplementary

Fig. 13). *CPT1* is the rate-limiting enzyme for the uptake of FFAs and their subsequent beta-oxidation in mitochondria[62]. *CPT1* might reduce lipotoxicity by promoting FFA consumption in hepatocytes. The newly identified *MGLL* and *CPT1* expression pattern in the yellowstripe goby liver may provide a new insight for future research on treating and preventing human NAFLD.

Yellowstripe goby had the highest *ABCA1* copy number among the species evaluated. *ABCA1* participates in the transport of CHOL from hepatocytes to the extracellular space[63]. Yellowstripe goby *ABCA1a-1* has similar functions to human *ABCA1*, whereas *ABCA1c* is an ancient subtype especially preserved in yellowstripe goby. Yellowstripe goby *ABCA1a-1* and *ABCA1c* are both highly expressed in the liver, which might increase the extrusion of free CHOL from hepatocytes and play an important role in reducing the lipotoxicity related to free CHOL.

Genome analysis revealed that the innate immunity gene *TLR23* is expanded in yellowstripe goby, with 15 copies, which is higher than that in most other fish, except round goby (40 copies)[8], Glacier lanternfish (*Benthosema glaciale*) (49 copies)[8], and European perch (*Perca fluviatilis*) (17 copies)[8]. In teleosts, TLR expansions might correlate with the survival and successful radiation of this lineage[64]. The expansion of *TLR23* in yellowstripe goby may help it to adapt to the complex environment, for instance, by reducing the amplification effect of the external environment on steatohepatitis. During the progression from simple NAFLD to NASH in humans, hepatocytes usually undergo multiple hits and eventually produce inflammatory reactions[63,65]. Pathogen- and damage-associated molecular patterns (PAMPs and DAMPs, respectively) are the main inducers of NASH inflammation. Both PAMPs and DAMPs can trigger inflammatory reactions in hepatocytes through TLRs[66]. However, the major function of *TLR23* is its involvement in identifying bacterial 23 S ribosomal RNA[67]. Therefore, the increased *TLR23* copy number in yellowstripe goby may be relevant in the inhibition of NASH caused by PAMPs, weakening the amplification effect of pathogens on NASH, and providing a stable innate immune environment in the liver to allow higher fat storage.

## Methods

### Genome sequencing and assembly

*DNA library construction and sequencing.* For sequencing, genomic DNA was isolated from the 7th generation of an inbred line. For whole-genome shotgun sequencing, four female full siblings were used: one to construct short-insert libraries of 270-bp and 800-bp, two for long-insert libraries of 20-kb, and one for a 40-kb long-insert library. Paired-end sequencing of the short-insert libraries was performed using the Illumina HiSeq 4000 system. After removing the adapter sequences, ambiguous and low-quality reads were filtered out using SOAPnuke[68] software, version 1.5.4 (https://github.com/BGI-flexlab/SOAPnuke) with the parameters: '-n 0.05 -l 7 -q 0.2 -d -i -Q 2'. The clean, high-quality data were used for genome assembly. Long-read libraries were sequenced using the Pacific Biosciences RSII system. After removing reads with a length of <500 bp or a score of <80, a total of 30.0 Gb of high-quality data was obtained. The study was approved by Institutional Animal Care and Use Committee (IACUC) of Guangdong Laboratory Animals Monitoring Institute, Guangzhou.

*Genome size estimation.* Filtered short reads were used to estimate the genome size and heterozygosity of yellowstripe goby by performing 17-mer analysis using the KmerFreq software, version 5.0 (https://github.com/fanagislab/kmerfreq). The genome size was estimated using the formula $G = \frac{N_{kmer}}{C_{k-mer}} = \frac{N_{read} \times (L-k+1)}{C_{k-mer}}$, where $G$ is the genome size, $N_{k-mer}$ and $N_{read}$ are the respective numbers of K-mers and reads, $C_{k-mer}$ is the average coverage depth of the K-mers, and $L$ and $K$ represent the read and K-mer lengths, respectively. A series of curve simulations was used to estimate the genome heterozygosity rate.

*Genome hybrid assembly.* The high-quality paired-end sequencing reads from the small-insert libraries were used to construct short, but accurate De Bruijn graph contigs using Platanus[69] (http://platanus.bio.titech.ac.jp/platanus-assembler, version 1.2.4) with the parameters: '-k 31 -s 10 -n 2 -c 3 -a 10.0 -u 0.2 -d 0.4 -t 16 -m 200'. The Celera Assembler PBcR pipeline (version 8.3rc2)[70] was used to correct the sequencing errors of the PacBio SMRT reads. The short, accurate contigs were then mapped to PacBio long reads to generate a hybrid assembly using

DBG2OLC[71] (https://github.com/yechengxi/DBG2OLC). The initial consensus sequences of DBG2OLC were polished to correct erroneous sequences due to the high error rates of the PacBio reads. SSPACE software (version 1.2.4)[72] was used for scaffolding the hybrid contigs (https://www.baseclear.com/services/bioinformatics/basetools/sspace-standard/). Finally, we used TrimDup, which is part of the Rabbit Genome Assembler (https://github.com/gigascience/rabbit-genome-assembler, version 2.6) with a percentage of 0.3 to remove redundant sequences.

**BUSCO evaluation.** We used BUSCO[73] software, version 3.0.2 to evaluate completeness and accuracy of the genome assembly. We selected actinopterygii_odb9 as the database, which contained 4584 highly conserved genes of fishes.

## Genome characterization

**Repeat detection.** We utilized both known and de novo methods for detecting repetitive DNA sequences. We used RepeatMasker (version 4.0.6)[74], RepeatProteinMask, and Tandem Repeats Finder (version 4.07)[75] to detect known transposable elements, transposable element-related proteins, and tandem repeats, respectively. In addition, we constructed a de novo repeat library using RepeatModeler (version 1.0.8) and LTR_FINDER (version 1.0.6)[76], and then we employed RepeatMasker to find de novo transposable elements.

**Gene-structure prediction.** To identify candidate protein-coding genes, we first aligned great blue-spotted mudskipper (*Boleophthalmus pectinirostris*), zebrafish (*Danio rerio*), and Japanese medaka (*Oryzias latipes*) protein-coding genes against the yellowstripe goby genome using TblastN with an *E* value of 1e-5. Then, we used GeneWise (version 2.4.1)[77] for precise alignments and gene-structure predictions. We used AUGUSTUS software (version 3.2.1)[78] and the GENSCAN[79] web server for ab initio gene-structure predictions. Furthermore, Illumina HiSeq RNA-Seq transcriptome data was used. HiSeq RNA-Seq reads from two liver tissues[37] were mapped on to the yellowstripe goby genome using Hisat2 (version 2.0.2)[80]. Finally, all the above data were combined to generate a comprehensive gene set using GLEAN[81].

**Gene function annotation.** Gene functions were annotated by aligning yellowstripe goby protein sequences to public databases (NT, NR, COG, KEGG, Swiss-Prot, and TrEMBL) using BlastP with an *E* value of 1e-5. InterProScan analysis was performed by running the ProDom, PRINTS, HAMAP, and Pfam applications. Gene Ontology annotations were extracted from the NR database using Blast2GO[82].

**Non-coding RNA predictions.** Non-coding RNAs (including microRNAs, ribosomal RNAs, small nuclear RNAs, and transfer RNAs) were predicted by comparing the yellowstripe goby genome against public libraries.

**Ks analysis.** For the Ks analysis, we first blasted the proteins sequences of three species (*M.chulae, Boleophthalmus pectinirostris, Oncorhynchus mykiss*) using blastp with themselves or between two species (Parameter: -m 8 -e 1e-5 -b 5 -v 5). The alignment was then processed by MCScanX to get collinear blocks (Parameter: -k 200 -g -2 -m 15 -s 5); each block contained no less than 5 gene pairs. Next, we connected gene pairs of each block into two supergenes and carried out sequence alignment with MUSCLE. Finally, we converted the protein sequences into nucleotides and carried out selective pressure analysis by PAML, and calculated Ks values of every block.

## Comparative genome analysis

**Gene family construction.** We assembled gene sets from 18 species (*Anolis carolinensis, B. pectinirostris, Cynoglossus semilaevis, D. rerio, Takifugu rubripes, Gadus morhua, Gallus gallus, Gasterosteus aculeatus, Homo sapiens, Larimichthys crocea, Mola mola, Notothenia coriiceps, Oreochromis niloticus, O. latipes, Tetraodon nigroviridis, Thunnus orientalis, Xenopus tropicalis*, and *Xiphophorus maculatus*), in addition to yellowstripe goby. All-to-all BlastP was performed using all protein sequences with *E* value of 1e-7. Hcluster_sg (https://github.com/douglasgscofield/hcluster) was used for protein clustering.

**Phylogenetic tree construction.** Using the clustered families, single-copy protein-encoding genes were extracted and multiple-sequence alignments were performed using MUSCLE (version 3.8.31)[83]. Corresponding coding sequence alignments were determined from protein alignments and were joined to form a 'supergene' for every species. We removed poorly aligned positions and divergent regions using Gblocks software (version 0.91b)[84] with default parameters, before constructing a phylogenetic tree using RAxML software (version 8.2.4)[85] with the GTRGAMMA model.

**Divergence time estimation.** The MCMCTree module from the PAML package was used to estimate the divergence time of yellowstripe goby from *B. pectinirostris* and other species. We selected several reference divergence times (marked by red dots in several branches) from the TimeTree database[86] (http://www.timetree.org/) to calibrate the divergence times for other nodes.

**Gene family expansion and extraction.** We used Computational Analysis of gene Family Evolution (CAFE) software[87], version 2.1 to analyze the changes in family sizes that occurred during the phylogenetic history. Prior to this analysis, we removed gene families with changes that were either too large (≥200) or too small (≤2) in size, as these could lead to wrong parameter estimations in CAFE.

**Gene copy-number scanning.** Protein sequences were downloaded from the NCBI or KEGG database. All-to-all BlastP was performed to assess the criteria of coverage and identity cutoffs. Then, these proteins were used to scan all genomes to determine gene copy numbers for each species.

## Genetic map construction

**RAD sequencing.** RAD sequencing libraries from the two parents and 225 F1-generation offspring were constructed and pooled into 22 final libraries with equal amounts of products, which were then sequenced on an Illumina HiSeq 4000 sequencing platform.

**RAD sequencing data analysis.** After removing adapters and low-quality bases, the clean reads were assigned to each individual based on specific barcodes and the *Eco*RI recognition site (GAATTC). Reads that did not contain a matching, unique barcode were discarded. All reads were aligned against the reference yellowstripe goby genome using BWA software (version 0.7.12)[88]. Single-nucleotide polymorphisms (SNPs) were detected using SAMtools software (version 1.2.1)[89] with the parameters: 'mpileup -g -d 100 -q 20 -Q 15', and bcftools software (version 1.3.1). Subsequently, we identified divergent SNP sites that differed between parents and sites that were heterozygous in either parent. Then, the basetypes of each individual offspring were extracted to construct a final basetype table.

**Linkage map construction.** SNP markers were filtered before they were used for linkage map construction by removing the following: (1) markers whose genotypes were the same between the parents or homozygous in both parents, (2) offspring samples in which <80% of the SNP sites were genotyped, (3) markers that could not be genotyped in ≥1% of the offspring samples, (4) markers with significantly distorted segregation ($P < 0.05$) in $x^2$ goodness-of-fit tests, and (5) redundant markers linkage disequilibrium <0.8. Paternal- and maternal-specific linkage maps were constructed using both JoinMap (version 4.0)[90] and LepMap2[91] with cross-pollinator population type codes and a logarithm of odds score limit of 20. After removing markers that came from the same sequence but were located on different linkage groups or with contradictory orders, an integrated sex-averaged linkage map was obtained.

## Sex chromosome identification
To identify sex chromosomes, we conducted gender identification on the F1 individuals using a microanatomy method, because of the tiny size of the samples. Of the 225 F1 individuals, 67 were identified as males, 72 were identified as females, and 86 were unidentifiable (Supplementary Table 18). We used mapQTL6 to identify the sex determination region by interval mapping (assigning a value of −1 for females and a value of 1 for males).

## Liver tissue sections of yellowstripe goby
Samples were anesthetized in MS222 (50 mg/L) and dissected to obtain the livers. Each liver was preserved in 4% neutral formaldehyde fixative for sectioning. The liver samples were dehydrated for routine pathology, fixed by paraffin embedding, and sliced using an automatic slicer (thickness, 4 μm). The sections were subjected to heating, pasting, drying, dewaxing, hematoxylin staining for 7 min, bluing with warm water for 1 min, and soaking in 1% hydrochloric acid alcohol for differentiation for approximately 60 s. Subsequently, they were subjected to eosin staining for 5 min, dehydration through an alcohol gradient, xylene hyalinization, neutral-resin sealing, observation under a microscope, and photographed.

## Determination of the hepatic lipid composition

**Experimental fish.** The experimental fish comprised a closed group of 12-month-old yellowstripe goby bred in our laboratory, totaling 300 individuals, which were divided into three groups.

**Hepatic lipid assays.** Total lipids, saturated fatty acids, monounsaturated fatty acids, polyunsaturated fatty acids, N-3, and N-6 were subjected to methyl esterification and analyzed using gas chromatography–mass spectrometry. Approximately 2 g of each sample was taken in a test tube, to which 1 mL of potassium hydroxide–methanol solution was added. Each test tube was sealed, placed in an oscillator to shake for approximately 30 min, and centrifuged at 5000 rpm in a high-speed centrifuge for approximately 5 min. The supernatants were used for testing.

After dehydration, total CHOL ester was quantitatively evaluated by sulfur, phosphorus, and iron indicator-spectrophotometry. Free CHOL ester was measured by staining with Coomassie brilliant blue. Glycerol ester and diglyceride were determined by spectrophotometry. The free and total glycerol contents after diglyceride hydrolysis were determined separately under different hydrolysis conditions. Total fatty acid contents were measured by acid–base titration. Diphosphatidylglycerol, lysolecithin, phosphatidylcholine,

phosphatidylethanolamine, phosphatidylserine, and sphingomyelin were measured by high-performance liquid chromatography.

**Validation of differentially expressed genes by quantitative real-time PCR.** Samples used for sex determination and lipid metabolism genes validation had been described in the Materials subsection under Transcriptome Sequencing (Supplementary Methods). Reverse transcription of total RNA was conducted following the manufacturer's protocol (Takara Bio, China). Quantitative real-time PCR amplification was performed on an ABI7500 system (Thermo, USA) using a TB Green kit (Takara Bio, China). The relative expression levels of target genes were calculated by the $2^{-\Delta\Delta Ct}$ method[38].

**Statistics and reproducibility.** Statistical analysis was performed by Student's $t$ test (between two groups) and one-way ANOVA (among three or more groups) using SPSS 17.0 software (SPSS Inc.). A value of $P$ value $< 0.05$ was used to indicate a significant difference.

**Reporting summary.** Further information on research design is available in the Nature Research Reporting Summary linked to this article.

## Data availability

The yellowstripe goby whole-genome project has been deposited in NCBI under project PRJNA598084. RAD sequencing reads have been deposited in the NCBI Sequence Read Archive under project PRJNA642226, and RNA-Seq sequencing reads have been deposited in the NCBI Sequence Read Archive under project PRJNA641222.

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

## Acknowledgements
We acknowledge Y. Zhang for their support on the yellowstripe goby genome project. We thank M. H. Wu, X, Q, Chen, S. Q. Lai, Z. T. Lin, J. Zeng, and X. Y. Lin for their assistance in sample collection and W. L. Wu for preparing the graphs. We acknowledge grant support from the National Key Technologies R & D Program of China (Grant No. 2015BAI09B05).

## Author contributions
R.H. initiated the yellowstripe goby genome project. R.H., L.C., Y.W., G.L., and Y.Z. conceived the study. L.C., Y.W., G.L., and Y.Z. wrote and revised the manuscript and the supplementary materials. L.C. and Y.W. conducted the genome and transcriptome analysis and performed the comparative genomic and genome evolution studies. G.L. and Y.Z. conducted the genome assembly, annotation, comparative genomics analysis, genetic map construction, and transcriptomic analysis. J.L., Z.M., L.Y., M.C., and H.Y. conducted the sample preparation, and gene validation. Z.Y., Z.D. and W.S. supervised the sequencing, assembly, and bioinformatics analysis. M.C. coordinated the project.

## Competing interests
The authors declare no competing interests.
