## [Peer Review File · Communications Biology]

Reviewers' comments:

Reviewer #1 (Remarks to the Author):

The authors generated a chromosome-level genome assembly for yellowstripe goby and identified a single sex determination region on chromosome 5. Differentially expressed genes associated with lipid accumulation in liver was identified. The authors also evaluated gene expansion which could explain the high-fat storage in liver of yellowstripe goby.

This study generates the first reference genome in the subfamily Gobionellinae which would be valuable for various studies of sex determination and liver high-fat storage in marine fishes. However, the authors should be more cautious on data collection and analysis steps, which is crucial to this project.

1. Line 31, please explain why it is important to do sex manipulation for yellowstripe goby breeding? Is there any report of growth vigor of a particular sex?
2. It would be helpful to list liver fat proportion of the other marine fish. Is NASH a problem with any other marine fish?
3. The author should include estimated genome size and heterozygosity based on the kmer analysis in the result. Is there any karyotype analysis of genome size and chromosome numbers?
4. Why did the author choose platanus to do the assembly, which is designed for species with high heterozygosity? It is not a good idea to use 4 individuals for genome assembly, especially when the heterozygosity is high. What is the sex of those fish? Base on the sex-linked SNP, it seems yellowstripe goby has a XX/XY sex determination system, is that true? If those four fish have different sex, it would be a problem for the assembly step.
5. 28.7X PacBio reads should allow a long-reads only assembly. Did the authors try Canu, FALCON or minimap2 to generate a genome backbone and run Pilon or other polish tools to get the reference assembly?
6. What is the size of anchored assembly with genetic linkage map?
7. What is the physical size of the sex QTL?
8. Is S247-888402 the highest SNP in figure 2? The author could try to add the stacked SNPs back after the QTL analysis to narrow down the sex determination region. Also In supplementary table 8, SNP positions should be included.
9. If yellowstripe goby has a XY system, then how could there're AA in male fish while all the females are GG for S247-888402? Did the author observe allele-specific expression of S247-888402 in male? Please discuss the mechanism as the SNP is in the intron.
10. To identify the sex determination gene, RNA should be extracted through gonad developing stage, please explain why 12-month fish are selected and on which day does the gonad show differentiation in yellowstripe goby.
11. In the abstract the authors mentioned four sex determination genes, but in the results and discussion section only two of them are mentioned. In supplementary table 10, Ube2a shows a much higher expression in males, why didn't the author think it is a candidate other than H2afy?
12. The authors should run RT-PCR to validate their findings for those key genes such as MSL3, H2AFY, MGLL and CPT1.
13. As liver fat accumulates between 10-60 days, it is confused why the end point (G2M) to be selected as the study group other than multiple time points during the period such as 10 days, 30 days, 60 days? At least a medium time point is better than the last one, which may already have a cooldown transcriptome profile.

In conclusion, a substantial revise is anticipated before the publication of this manuscript.

Reviewer #2 (Remarks to the Author):

The authors present a new whole genome sequence for the yellowstripe goby. Gobies are a highly

speciose family of marine fishes so increased coverage of the lineage is likely to be interesting. The authors argue that gobies are interesting because they could be used as a marine model species, which they argue are rare, and because of their diverse reproductive modes and to the high fat content of their liver which causes health problems in humans. The text is largely clear and to the point, but lacks wider relevance of the study, apart from human disease. For example, the sex determination section is not set into a wider context and there is no mention of recent high-profile papers on the evolution of fat regulation in teleosts, e.g. the stickleback paper in Science from Kitano's group (DOI: 10.1126/science.aau5656) is not mentioned but could help to illuminate the wider relevance as could the work from Rasmus Nielsen's group on adaptation to a high fat diet in human inhabitants of Greenland (DOI: 10.1126/science.aab2319).

I find the justification is a little short on detail. For instances, they do not specify what sort of marine models are lacking and why having one may be of interest, over and above the standard teleost model species, zebrafish, medaka and pufferfish. One of the standard model fish species, the pufferfish, are in fact marine. Why it is interesting to sequence a species in a lineage that has diverse reproductive modalities is not specified in the text.

Although most of the results section is given over to details of the expression of liver genes, there is no section relating to this analysis in the methods section. Particularly, it is not clear how differential expression is assessed, the authors repeatedly state that fat pathway genes are upregulated in the yellowstripe goby liver, but it is not clear what this is in comparison to as, although there are multiple tissues presented in figure 4b, there is no detail about this in the methods. Even with the addition of other tissues it is largely unclear if the upregulation of genes with fat related functions or within fat related pathways is simply what you expect for the liver and how the authors decide if this is particularly exaggerated in yellowstripe goby. Without this information it is not possible to properly assess the reliability of large parts of the manuscript.

Specific comments:

The manuscript is very abbreviation heavy, please review whether these are always required. It would be helpful to the reader to have fewer to deal with if possible. Please try and remove any that are not strictly standard (e.g. gene symbols are helpful, but do you need to shorten lipid droplets to LDs?).

Line 25: what is meant by "fecundity strategy"? Is "reproductive strategy" better here since there is no other information on fecundity presented in the manuscript? However, it is still not clear what about this would make them globally distributed.

Line 27: "bisexualism" is this the right word?

Line 27: "bidirectional" this reads like this is possible within a species, do you mean that there is both protoandrous and protogynous sex change?

Line 31: what is meant by "sex breeding" this is not clear

Line 48: Is this really true, what about pufferfish? Stickleback is essentially a marine fish which has undergone repeated freshwater invasions. A model for what purpose and why would a marine model be required?

Line 53: Not clear what this is, can you add something that makes it clear what this standard is for? What sort of biological research is it intended as a model in?

Line 77: quote the BUSCO score?

Line 97: was anything done to assess the quality of the resultant chromosome assembly -

comparison to synteny in another species with a high-quality reference? Figure 6 of the supplementary shows synteny to zebrafish, but this is not very well conserved. Is this because zebrafish is too distant or because the linkage map did not resolve the structure well? Is there another genome that you could compare to? There is no discussion in the manuscript about what this low synteny means.

Line 106: representative of what?

Line 119: "However, this peak continued into the following peak."

Fragment please revise, particularly please make it clear which peak you are referring to.

Line 121: It is not clear what this means? What is meant by "higher-level phylogeny" and what does this have to do with orthologue identification? Why is the one to one relationship low? Is it because the zebrafish has retained relatively more retained duplicates from the 3R vertebrate duplication? And so many genes have a one to two relationship when compared to zebrafish? Or has this 1:2 relationship hampered the identification of orthology relationships? Did you try comparing to medaka, in this comparison you may find more 1:1, or even 2:1 relationships, than in comparison with zebrafish.

Line 127: using what test?

Line 128: signal of what, you need to say that this is a LOD score from linkage mapping.

Line 137 and 141 and throughout: "spermary" I think testis is a more well used/standard term for teleosts.

Line 157: what is meant by "by heredity"

Line 160: "largely exceeds" or exceeds by a large degree?

Line 170: and throughout these gene expression sections: upregulated compared to what?

Line 171: "discarded" Was there a reason to suspect the data were weak other than a lack of correlation with earlier developmental stages? Was the sample size too small? Please explain especially since the data are used later in the manuscript so these data were not discarded completely.

Line 197: It is not clear if this is what is normally expected for the liver or something particular to the high lipid content of yellowstripe goby livers.

Line 206: remove repeated "expression" and change high to highly

Line 208: remove "of"

Line 226: up-regulated compared to what?

Line 232: earlier stated that this G3M data was discarded. Please reword the preceding statement to say it was not included in that analysis and state why.

Line 239: in this in the yellowstripe goby?

Line 244: strange phrasing, what is meant by older evolutionary status. Is this the ancestral copy? Which whole genome duplication is it likely derived from?

Line 253: all species sequenced in this study?

Line 265: change benefit to be beneficial

Line 278: no rationale given as to why might H2AFY be related to sex determination

Line 286: A brief statement of the wider relevance of this study and any explanation as to why gobies would need such high lipids would be helpful here.

Line 364: were the Iso-seq reads also from just liver?

Line 367-368: but were the transcripts really just from liver? This would mean many genes that are not expressed in the liver would be removed. Could this explain the low overlap in genes with other taxa?

Line 379: please restate here which 3 species you refer to.

Line 416: each of the 18 species?

Line 438: less than or more than? What is a LD score - do you mean simply LD? How was LD calculated?

We appreciate for Editors/Reviewers' warm work earnestly during this special time of the COVID-19 pandemic. We would like to thank the reviewers for their thoughtful review and comments on the manuscript, entitled “**Whole-genome sequencing reveals sex determination and liver high-fat storage mechanisms of yellowstripe goby (*Mugilogobius chulae*)**”. Those comments are all valuable and very helpful for revising and improving our paper, as well as the important guiding significance to our researches. We have studied comments carefully and have made correction which we hope meet with approval. **Revised portion are marked in red in the paper.** The point to point responds to the reviewer's comments are listed as following:

Reviewers' comments:

Reviewer #1 (Remarks to the Author):

The authors generated a chromosome-level genome assembly for yellowstripe goby and identified a single sex determination region on chromosome 5. Differentially expressed genes associated with lipid accumulation in liver was identified. The authors also evaluated gene expansion which could explain the high-fat storage in liver of yellowstripe goby.

This study generates the first reference genome in the subfamily Gobionellinae which would be valuable for various studies of sex determination and liver high-fat storage in marine fishes. However, the authors should be more cautious on data collection and analysis steps, which is crucial to this project.

1. Line 31, please explain why it is important to do sex manipulation for yellowstripe goby breeding? Is there any report of growth vigor of a particular sex?

Response: Thank you for your patient review and valuable suggestions. Considering that gobiid fish is one of the most important biological groups in the marine ecosystem, understanding its sex determination mechanism is of great significance for revealing the adaptive strategies underlying their global distribution and providing invaluable insights on the sex determination evolution in teleost.. In our previous research, we found that the growth speed of male yellowstrip goby was faster than female ^[1].

We have modified this paragraph for better understanding. Please see the details in the following:

“Considering that gobiid fish is one of the most important biological groups in the marine ecosystem, understanding its sex determination mechanism is of great significance for revealing the adaptive strategies

underlying their global distribution and providing invaluable insights on the sex determination evolution in teleost.”

Reference

[1] LI, J. J., et al. Analysis on morphology and growth characteristics of *Mugilogobius chulae*. Lab Anim Comparative Med. (2012), 32: 224-240.

2. It would be helpful to list liver fat proportion of the other marine fish. Is NASH a problem with any other marine fish?

Response: As suggested by the reviewer, we had added one table about the liver fat proportion of the other marine fish in supplementary materials (supplementary Table 12). NASH was one of serious hepatic disease in most farmed fish, however, in the majority of gobiids, large quantities of neutral fat are stored in the liver as an energy storage and no NASH symptoms are discovered in liver, suggesting the existence of a specialized mechanism for maintaining a balance between high-fat storage and inflammation in the liver in these fishes.

Supplementary Table 12 The lipid content in marine fishes

Species	Total lipids (% of livers)	References
Yellowstripe goby (M.chulae)	77	This study
European sea bass (Dicentrarchus labrax)	33.81	1
Cod (Gadus morhua L.)	38.30	2
Red sea bream (Pagrus major)	21.83	3
Amberjack (Seriola dumerili)	33.32	3
Striped jack (Caranx delicatissimus)	53.26	3

References

1. dos Santos, J., Burkow, I. C., & Jobling, M.. Patterns of growth and lipid deposition in cod (*Gadus morhua* L.) fed natural prey and fish-based feeds. *Aquaculture*, 110(2), 173-189 (1993).
2. Valente, L. M. P., Bandarra, N. M., Figueiredo-Silva, A. C., Cordeiro, A. R., Simoes, R. M., & Nunes, M. L.. Influence of conjugated linoleic acid on growth, lipid composition and hepatic lipogenesis in juvenile European sea bass (*Dicentrarchus labrax*). *Aquaculture*, 267(1-4), 225-235 (2007).
3. Ando S , Mori Y , Nakamura K , et al. Characteristics of Lipid Accumulation Types in Five Species of Fish.[J]. *Nippon Suisan Gakkaishi*, 1993, 59(9):1559-1564.

3. The author should include estimated genome size and heterozygosity based on the kmer analysis in the result. Is there any karyotype analysis of genome size and chromosome numbers?

Response: Thank you for your valuable suggestions. As suggested by the reviewer, we have added the genome size and heterozygosity based on the kmer analysis in this manuscript. According to our previous research, the chromosome numbers of yellowstripe goby were 22^[1]. The genome size of this fish was first reported in this manuscript.

Please see the details in the following:

“Using 65.29 Gb of sequencing data from the HiSeq platform for 17-mer analysis, the heterozygosity of the yellowstripe goby genome was calculated to be 1.2% (Supplementary Fig. 3).”

Supplementary Fig. 3 Genome size estimation (k-mer=17)

Reference

[1] Xiao-Qu, Chen , H. Ren , and L. I. Jian-Jun . Study on the karyotype of *Mugilogobius chulae*. *Journal of Tropical Oceanography* 32 : 88-95, (2013).
 Doi: 10.3969/j.issn.1009-5470.2013.06.013

4. Why did the author choose platanus to do the assembly, which is designed for species with high heterozygosity? It is not a good idea to use 4 individuals for genome assembly, especially when the heterozygosity is high. What is the sex of those fish? Base on the sex-linked SNP, it seems yellowstripe goby has a XX/XY sex determination system, is that true? If those four fish have different sex, it would be a problem for the assembly step.

Response: Yes, platanus is efficient de novo assembly software for highly heterozygous genomes, we chose it because the heterozygosity of yellowstripe goby reached 1% based on Kmer analysis. We also tried SOAPdenovo, but Platanus performed better.

The four individuals were all female and from the same parents.

DNA for constructing mate-pair library was needed upto 40 ug per library, and one individual is too small to extract enough DNA for constructing multiple libraries. So we had to use 4 individuals for genome assembly. We guaranteed that all the small inserted size (270bp and 800bp) pair-end libraries came from the same individual, which ensures the accuracy of the contigs. Based on the existing data in this manuscript, we temporarily unable to confirm the sex determination system of yellowstripe goby was XY or not.

5. 28.7X PacBio reads should allow a long-reads only assembly. Did the authors try Canu, FALCON or minimap2 to generate a genome backbone and run Pilon or other polish tools to get the reference assembly?

Response: Actually we had tried to do a long-reads only assembly using Canu, however its performance was poorer than that of DBG2OLC, which took a hybrid assembly strategy.

6. What is the size of anchored assembly with genetic linkage map?

Response: The size of anchored assembly with genetic linkage map is 922 Mb, ~92% of the genome size. We have modified the manuscript.

Please see the details in the following:

“Using the genetic linkage map, we anchored the assembly to 22 chromosomes (Fig. 1a), containing 1,065 scaffolds and 922 Mb (92%) of the total length of the assembled sequences”

7. What is the physical size of the sex QTL?

Response: A 20.67-cM sex determination region was discovered on chromosome 5, and its physical size is ~49.2 Mb.

We have modified the manuscript. Please see the details in the following:

“Notably, we detected a strong signal (log of the odds score =12.5) with a 20.67-cM-broad peak on chromosome 5 (Fig. 2), representing 49.2 Mb physical size.”

8. Is S247-888402 the highest SNP in figure 2? The author could try to add the stacked SNPs back after the QTL analysis to narrow down the sex determination region. Also In supplementary table 8, SNP positions should be included.

Response: NO. S247-1213950 is the highest in QTLs analysis. But S247-888402 is nearby S247-1213950 and is only 325 Kb between each other. Because of the quantity of F1 individuals which the sex were identified, the mapQTL6 soft has reached the limit to narrow down the sex determination region. We have been modified the supplementary table 8.

Please see the details in the following:

Supplementary Table 8 Significantly associated SNPs[Ⓢ]

SNP ID [Ⓢ]	type(♀) [Ⓢ]	type(♂) [Ⓢ]	SNP position [Ⓢ]
s247-672252 [Ⓢ]	G [Ⓢ]	R [Ⓢ]	scaffold247:672252 [Ⓢ]
s247-764174 [Ⓢ]	G [Ⓢ]	R [Ⓢ]	scaffold247:764174 [Ⓢ]
s247-842340 [Ⓢ]	C [Ⓢ]	Y [Ⓢ]	scaffold247:842340 [Ⓢ]
s247-842371 [Ⓢ]	C [Ⓢ]	Y [Ⓢ]	scaffold247:842371 [Ⓢ]
s247-849608 [Ⓢ]	A [Ⓢ]	R [Ⓢ]	scaffold247:849608 [Ⓢ]
s247-849806 [Ⓢ]	G [Ⓢ]	R [Ⓢ]	scaffold247:849806 [Ⓢ]
s247-888402 [Ⓢ]	G [Ⓢ]	R [Ⓢ]	scaffold247:888402 [Ⓢ]
s247-1213950 [Ⓢ]	C [Ⓢ]	S [Ⓢ]	scaffold247:1213950 [Ⓢ]
s247-1359738 [Ⓢ]	A [Ⓢ]	M [Ⓢ]	scaffold247:1359738 [Ⓢ]
s247-1831678 [Ⓢ]	C [Ⓢ]	Y [Ⓢ]	scaffold247:1831678 [Ⓢ]
s247-1921993 [Ⓢ]	G [Ⓢ]	R [Ⓢ]	scaffold247:1921993 [Ⓢ]
s247-1921995 [Ⓢ]	G [Ⓢ]	R [Ⓢ]	scaffold247:1921995 [Ⓢ]

9. If yellowstripe goby has a XY system, then how could there're AA in male fish while all the females are GG for S247-888402? Did the author observe allele-specific expression of S247-888402 in male? Please discuss the mechanism as the SNP is in the intron.

Response: Based on the existing data in this manuscript, we could not confirm the sex determination system of yellowstripe goby was XY or not. Meanwhile, we hadn't found allosome in yellowstripe goby based on karyotype analysis ^[1]. We also validated the expression of this gene (*GALNT10-like*) in different tissues of female and male individuals by quantitative real-time PCR, results showed that *GALNT10-like* gene was only high expressed in ovarian tissue (supplementary figure 7c). The relationship between *GALNT10-like* gene and its SNP of S247-888402 with sex determination might be worthy of further investigation. This new figure have added in supplementary materials (supplementary figure

7c) and some discussion about the mechanism of the SNP in the intron have also been added in discussion of the manuscript.

Please see the details in the following:

“*GALNT10-like* participates in the synthesis of mucin-type oligosaccharides⁴⁷. We found a SNP that was located in the second intron of this gene to be homozygous (GG) in all female fish. Quantitative real-time PCR validation of *GLANT10-like* in different tissues of female and male individuals also revealed that *GLANT10-like* was only highly expressed in ovary. Most fish eggs are surrounded by a glycoproteinaceous structure^{48,49}, called an egg envelope, which provides the embryo with physical protection. *GLANT10-like* might be participating the formation of the egg envelope and filament adhesion apparatus of yellowstripe goby. Sequence analysis showed that locus S247-888402 was located in the second intron of the *GALNT10-like* gene. The relationship between *GALNT10-like* gene and its SNP at S247-888402 with respect to sex determination might be worthy of further investigation.”

Fig7c. Relative expression of *GALNT10-like* in different tissues of female and male individuals. Means not sharing a common letter are significantly different at $p < 0.05$, as assessed using one-way ANOVA followed by the Dunnett’s test. (FOV~FHE: female fish, MTE~MHE: male fish. OV: vary, BR: rain, EY: eye, GI: gill, INT: intestine, LI: liver, MU: muscle, SK: skin, SP: spleen, HE: heart.)

Reference

- [1] Xiao-Qu, Chen , H. Ren , and L. I. Jian-Jun . (2013). Study on the karyotype of *Mugilogobius chulae*. *Journal of Tropical Oceanography* 32 : 88-95.
Doi: 10.3969/j.issn.1009-5470.2013.06.013

10. To identify the sex determination gene, RNA should be extracted through gonad developing stage, please explain why 12-month fish are selected and on which day does the gonad show differentiation in yellowstripe goby.

Response: Sexual maturity of yellowstripe goby is reached at about three months of age. However, because of the smaller body size of yellowstripe goby, it's difficult to distinguish female or male from morphology before six months of age. Meanwhile, the best breeding period of yellowstripe goby is between the six and eighteen months of age. In order to gather female and male gonad tissues more convenient, we selected 12 months of age fish for transcriptome sequencing. We also validated the expression of these sex determination genes by RT-QPCR using different developmental stages gonad tissues (Supplementary Figure 7).

According to our previous study, male and female gonad could be distinguished by histological observation just after 50 days and 70 days old (unpublished), respectively. Based on the research of zebrafish, sex determination is expected to occur before 10 dpf (days post-fertilization)^[1]. So, we speculate that the gonad differentiation time of yellowstripe may be similar with zebrafish.

Reference

[1] Orban, L., Sreenivasan, R., & Olsson, P. E. (2009). Long and winding roads: testis differentiation in zebrafish. *Molecular and cellular endocrinology*, 312(1-2), 35-41.

11. In the abstract the authors mentioned four sex determination genes, but in the results and discussion section only two of them are mentioned. In supplementary table 10, Ube2a shows a much higher expression in males, why didn't the author think it is a candidate other than H2afy?

Response: Thank you for your valuable suggestions. As suggested by the reviewer, we have modified the abstract, results and discussion. H2AFY and UBE2a were both located in sex determination region on chromosome 5. According to previous research, H2AFY might participate maintaining X-chromosome inactivation^[1] and UBE2a was associated with cognitive disability^[2]. Although the gene of UBE2a showed a much higher expression in males than H2AFY, there was still lack adequate evidence to confirm the relationship between UBE2a and sex determination. So we have deleted the information about UBE2a.

References

- [1] Hernández-Muñoz, I., Lund, A. H., Van Der Stoep, P., Boutsma, E., Muijers, I., Verhoeven, E., ... & Van Lohuizen, M. (2005). Stable X chromosome inactivation involves the PRC1 Polycomb complex and requires histone MACROH2A1 and the CULLIN3/SPOP ubiquitin E3 ligase. *Proceedings of the National Academy of Sciences*, 102(21), 7635-7640.
- [2] De Leeuw, N., Bulk, S., Green, A., Jaekle-Santos, L., Baker, L. A., Zinn, A. R., ... & Van Bokhoven, H. (2010). UBE2A deficiency syndrome: Mild to severe intellectual disability accompanied by seizures, absent speech, urogenital, and skin anomalies in male patients. *American Journal of Medical Genetics Part A*, 152(12), 3084-3090.

12. The authors should run RT-PCR to validate their findings for those key genes such as MSL3, H2AFY, MGLL and CPT1.

Response: Thank you for your valuable suggestions. As suggested by the reviewer, we have validated all the sex determination and lipid metabolism relative genes by RT-QPCR (Supplementary Figure 7 and Supplementary Figure 13). We also added these data in our manuscript.

13. As liver fat accumulates between 10-60 days, it is confused why the end point (G2M) to be selected as the study group other than multiple time points during the period such as 10 days, 30 days, 60 days? At least a medium time point is better than the last one, which may already have a cooldown transcriptome profile.

Response: We are sorry for the insufficient statement about the end point of liver fat accumulation. Histological observations of liver showed that fat was rapidly accumulated in hepatocyte between 10 and 60 days of age and was full filled with lipid droplet at 3 months of age. Because of the small body size of yellowstripe goby, the liver tissue in 10 days and 30 days of age were hardly acquired. In order to obtain enough liver tissue for transcriptome sequencing, we selected 2 months of age fish liver as the rapid accumulation period group, and 3 months of age fish as the control group. We also confirmed the expression of lipid metabolism related genes by RT-QPCR using the liver of 2 months and 3 months of age (Supplementary Figure 13).

Reviewer #2 (Remarks to the Author):

1. The authors present a new whole genome sequence for the yellowstripe goby. Gobies are a highly speciose family of marine fishes so increased coverage of the lineage is likely to be interesting. The authors argue that gobies are interesting because they could be used as a marine model species, which they argue are rare, and because of their diverse reproductive modes and to the high fat content of their liver which causes health problems in humans. The text is largely clear and to the point, but lacks wider relevance of the study, apart from human disease. For example, the sex determination section is not set into a wider context and there is no mention of recent high-profile papers on the evolution of fat regulation in teleosts, e.g. the stickleback paper in Science from Kitano's group (DOI: 10.1126/science.aau5656) is not mentioned but could help to illuminate the wider relevance as could the work from Rasmus Nielsen's group on adaptation to a high fat diet in human inhabitants of Greenland (DOI: 10.1126/science.aab2319).

Response: We appreciate the reviewer for this important comment. We have added some recent high-profile papers about sex determination and evolution of fat regulation in teleosts in the introduction of this manuscript.

Please see the details in the following:

“Compared with terrestrial vertebrates, sex determination mechanisms in fish have exhibited a

high degree of plasticity and complexity^{1,2}, which may be related to genetic or environmental factors or both. So far, several sex determination systems have been identified in fishes, including male-heterogametic gonochorism (XY)³, female-heterogametic gonochorism (ZW)⁴, hermaphroditism⁵, and environmental dependency⁶. However, the reasons for so many diverse sex determination mechanisms to have evolved and the key factor for transition of different sex determination mechanisms remain unknown⁷. Especially, in gobiid fish, one of the largest fish families, comprising more than 2000 species, there is little information about the sex determination mechanism of gobiids have been discovered, no sex determination genes or sex determination region have been identified to date⁸⁻¹⁰. Considering that gobiid fish is one of the most important biological groups in the marine ecosystem, understanding its sex determination mechanism is of great significance for revealing the adaptive strategies underlying their global distribution and providing invaluable insights on the sex determination evolution in teleost.

Lipids and their constituent fatty acids are major organic constituents from worm (*Caenorhabditis elegans*) to humans, and the ability to store fats is conserved^{11,12}. Interestingly, starting with the primitive teleosts (jawless vertebrates such as lampreys), the lipid-storing cells have evolved into a tissue that has distinct functions underneath the skin¹³, while the type and sites of fat storage is species-specific in fish and depend on the nutritional state, life-stage, and the physiological state^{14,15}. Most fishes, such as zebrafish (*Danio rerio*)¹⁴ and cavefish (*Astyanax mexicanus*)¹⁶, show deposition of lipids [mainly triglycerides (TAG)] in mesentery and viscera. Surprisingly, in majority of the gobiids, the lipids are only stored in the liver¹⁷⁻²⁰. This phenomenon is also exhibited in some other marine fishes, such as puffer (*Takifugu rubripes*) and cultured flounder (*Paralichthys olivaceus*)²¹. The special lipid deposition organ in these fish might be an evolutionary adaptation to cope with the typical living environment, such as rapid energy mobilization, migration, or benthic adaptation. This is similar to the situation observed with Inuit, who display genetic and physiological adaptations to a diet rich in PUFAs²² and stickleback lineages (*Gasterosteus aculeatus* species complex), which have evolved different copy numbers of docosahexaenoic acid biosynthesis-related genes to realize freshwater colonization and radiation²³.

2. I find the justification is a little short on detail. For instances, they do not specify what sort of marine models are lacking and why having one may be of interest, over and above the standard teleost model species, zebrafish, medaka and pufferfish. One of the standard model fish species, the pufferfish, are in fact marine. Why it is interesting to sequence a species in a lineage that has diverse reproductive modalities is not specified in the text.

Response: Thank you for your valuable and thoughtful comments. As suggested by the reviewer, we have enriched the details about the current studies of standard teleost model species and the reason for the choice of yellowstripe goby.

Please see the details in the following:

“Gobiids (Teleostei, Gobiidae), commonly known as gobies, are a diverse and fascinating group with worldwide distribution²⁷, and is one of the most diverse families of vertebrates on earth²⁸. Therefore, Gobies represent a potential excellent model for experimental evolutionary adaptation studies. Unfortunately, neither the genome sequencing and phylogenetic relationships of many groups of gobies nor laboratory breeding and rearing methods are resolved. Only a few gobiids genomes have been sequenced, such as of round goby (Gobiidae: *Neogobius melanostomus*)⁸ and mudskipper¹⁰. Currently, only a few small fish species [e.g. zebrafish²⁹; Japanese medaka (*Oryzias latipes*)³⁰, platyfish (*Xiphophorus maculatus*)³¹] have been widely used in laboratory; however, the large majority of these inhabit in freshwater environments. Laboratory models of marine species are limited to the three-spined stickleback species (*Gasterosteus aculeatus*)³², pufferfish (*Takifugu rubripes*)³³ and the Atlantic silverside (*Menidia menidia*)³⁴. Considering the specific needs of laboratory animal, such as convenient for indoor large-scale cultivation and controlled year-round spawning and sexual maturation, it is necessary to develop a representative marine fish to supplement the marine model fish species.”

3. Although most of the results section is given over to details of the expression of liver genes, there is no section relating to this analysis in the methods section. Particularly, it is not clear how differential expression is assessed, the authors repeatedly state that fat pathway genes are upregulated in the yellowstripe goby liver, but it is not clear what this is in comparison to as, although there are multiple tissues presented in figure 4b, there is no detail about this in the methods. Even with the addition of other tissues it is largely unclear if the upregulation of genes

with fat related functions or within fat related pathways is simply what you expect for the liver and how the authors decide if this is particularly exaggerated in yellowstripe goby. Without this information it is not possible to properly assess the reliability of large parts of the manuscript.

Response: We fully agree with the reviewer. We have validated all the sex determination and lipid metabolism relative genes by RT-QPCR. We also added these data in this manuscript (Supplementary Figure 7, Supplementary Figure 13).

Specific comments:

4. The manuscript is very abbreviation heavy, please review whether these are always required. It would be helpful to the reader to have fewer to deal with if possible. Please try and remove any that are not strictly standard (e.g. gene symbols are helpful, but do you need to shorten lipid droplets to LDs?).

Response: We appreciate the reviewer for this important comment. As suggested by the reviewer, we have enriched the introduction and discussion. We also removed the unnecessary abbreviations (e.g. LDs...).

5. Line 25: what is meant by “fecundity strategy”? Is “reproductive strategy” better here since there is no other information on fecundity

presented in the manuscript? However, it is still not clear what about this would make them globally distributed.

Response: We have replaced the word “fecundity” by “reproductive” and modified this sentence for more readable.

6. Line 27: “bisexualism” is this the right word?

Response: We have replaced this word by “gonochorism” .

Please see the details in the following:

“So far, several sex determination systems have been identified in fishes, including male-heterogametic gonochorism (XY)³, female-heterogametic gonochorism (ZW)⁴, hermaphroditism⁵, and environmental dependency⁶.”

7. Line 27: “bidirectional” this reads like this is possible within a species, do you mean that there is both protoandrous and protogynous sex change?

Response: Thank you for your valuable comments. We have amended this sentence.

Please see the details in the following:

“So far, several sex determination systems have been identified in fishes, including male-heterogametic gonochorism (XY)³, female-heterogametic gonochorism (ZW)⁴, hermaphroditism⁵, and environmental dependency⁶.”

8. Line 31: what is meant by “sex breeding” this is not clear

Response: As suggested by reviewer, we have modified this paragraph.

Please see the details in the following:

“Considering that gobiid fish is one of the most important biological groups in the marine ecosystem, understanding its sex determination mechanism is of great significance for revealing the adaptive strategies underlying their global distribution and providing invaluable insights on the sex determination evolution in teleost.”

9. Line 48: Is this really true, what about pufferfish? Stickleback is essentially a marine fish which has undergone repeated freshwater invasions. A model for what purpose and why would a marine model be required?

Response: We have revised this paragraph to describe the difference between yellowstripe goby and other standard model fish (such as pufferfish and stickleback).

Please see the details in the following:

“Gobiids (Teleostei, Gobiidae), commonly known as gobies, are a diverse and fascinating group with worldwide distribution²⁷, and is one of the most diverse families of vertebrates on earth²⁸. Therefore, Gobies represent a potential excellent model for experimental evolutionary adaptation studies. Unfortunately, neither the genome sequencing and phylogenetic relationships of many groups of gobies nor laboratory breeding and rearing methods are resolved. Only a few gobiids genomes have been sequenced, such as of round goby (Gobiidae: *Neogobius melanostomus*)⁸ and mudskipper¹⁰. Currently, only a few small fish species [e.g. zebrafish²⁹; Japanese medaka (*Oryzias latipes*)³⁰, platyfish (*Xiphophorus maculatus*)³¹] have been widely used in laboratory;

however, the large majority of these inhabit in freshwater environments. Laboratory models of marine species are limited to the three-spined stickleback species (*Gasterosteus aculeatus*)³², pufferfish (*Takifugu rubripes*)³³ and the Atlantic silverside (*Menidia menidia*)³⁴. Considering the specific needs of laboratory animal, such as convenient for indoor large-scale cultivation and controlled year-round spawning and sexual maturation, it is necessary to develop a representative marine fish to supplement the marine model fish species. Yellowstripe goby (*Mugilogobius chulae*) is a representative fish of the Gobiidae family that is widely distributed along the western Pacific coast³⁵. This species has a moderate body size (adult body length, 3–5 cm), a short sexual maturity period, strong reproductive capacity, a short spawning interval, annual reproduction, easy indoor rearing, and easy genetic manipulation³⁶.”

10. Line 53: Not clear what this is, can you add something that makes it clear what this standard is for? What sort of biological research is it intended as a model in?

Response: We are very sorry for our inaccurate statement in this paragraph. We have added some details in this paragraph.

Please see the details in the following:

“In addition, a Chinese national quality-control standard, including genetic, microorganism, parasite, nutrition, and environment quality control, has been established for yellowstripe goby (draft national standard no. 20091329-T-469)³⁷.”

11. Line 77: quote the BUSCO score?

Response: As suggested by the reviewer, we have added the BUSCO score in the manuscript.

Please see the details in the following:

“BUSCO evaluation revealed that the assembled genome contained 87% of the known fish orthologous genes (Supplementary Table 2).”

12. Line 97: was anything done to assess the quality of the resultant chromosome assembly - comparison to synteny in another species with a high-quality reference? Figure 6 of the supplementary shows synteny to zebrafish, but this is not very well conserved. Is this because zebrafish is too distant or because the linkage map did not resolve the structure well? Is there another genome that you could compare to? There is no discussion in the manuscript about what this low synteny means.

Response: As suggested by reviewer, we compared yellowstripe goby with medaka, and they showed a more conserved collinearity than zebrafish (Figure 1a).

Please see the details in the following:

Figure 1a Maps of the 22 Yellowstripe goby chromosomes and of the 24 Japanese medaka chromosomes based on the positions of 11756 orthologous pairs

13. Line 106: representative of what?

Response: In addition to yellowstripe goby, 14 fishes and 4 representative vertebrates (human, chicken, lizard and toad) were selected for phylogenetic analysis (Fig. 1b).

14. Line 119: “However, this peak continued into the following peak.”

Fragment please revise, particularly please make it clear which peak you are referring to.

Response: This peak means divergence time between yellowstripe goby and mudskipper (the peak of the yellow line). We have modified this sentence.

Please see the details in the following:

“However, the peak with yellow line (Fig. 1d) continued into another peak with green line (Fig. 1d). Yellowstripe goby has undergone only three WGD events.”

15. Line 121: It is not clear what this means? What is mean by “higher-level phylogeny” and what does this have to do with orthologue identification? Why is the one to one relationship low? Is it because the zebrafish has retained relatively more retained duplicates from the 3R vertebrate duplication? And so many genes have a one to two relationship

when compared to zebrafish? Or has this 1:2 relationship hampered the identification of orthology relationships? Did you try comparing to medaka, in this comparison you may find more 1:1, or even 2:1 relationships, than in comparison with zebrafish.

Response: As suggested by reviewer, we compared the genome of yellowstripe goby with medaka, and found a more conserved collinearity of 1:1 relationship between them (Figure 1a).

16. Line 127: using what test?

Response: We used mapQTL6 to identify the sex determination region by interval mapping. Please see the details in section of “methods – sex chromosome identification”.

17. Line 128: signal of what, you need to say that this is a LOD score from linkage mapping.

Response: As suggested by the reviewer, we have added the LOD score in the manuscript. The LOD score is 12.5.

Please see the details in the following:

“Notably, we detected a strong signal (**log of the odds score =12.5**) with a 20.67-cM-broad peak on chromosome 5 (Fig. 2), representing 49.2 Mb physical size.”

18. Line 137 and 141 and throughout: “spermary” I think testis is a more well used/standard term for teleosts.

Response: We have accepted the reviewer’s suggestion, replaced all “spermary” by “testis”.

19. Line 157: what is meant by “by heredity”?

Response: According to the present study, we had found that the liver high-fat deposition was caused by genetic factors. We have modified this sentence.

Please see the details in the following:

“Histological observation of the liver of yellowstripe goby showed that liver high-fat deposition was present by genetic factors,

20. Line 160: “largely exceeds” or exceeds by a large degree?

Response: As suggested by the reviewer, we have replaced “largely exceeds” by “exceeds by a large degree”.

21. Line 170: and throughout these gene expression sections: upregulated compared to what?

Response: G3M was the control group. We have added some descriptions about the control group.

Please see the details in the following:

“Transcriptome analysis of livers from 2-month-old (G2M) and 3-month-old (G3M) fish revealed that lipid synthetic genes in the G2M group were globally up-regulated compared with that in the G3M group.....”

22. Line 171: “discarded” Was there a reason to suspect the data were weak other than a lack of correlation with earlier developmental stages? Was the sample size too small? Please explain especially since the data are used later in the manuscript so these data were not discarded completely.

Response: We are sorry for our careless mistake in the statement. In fact, G3M-2 was discarded, because of the low correlation of G3M-2 compared with G3M-1 and G3M-3 (Supplementary Fig. 12). The mistake has been corrected in the manuscript.

Please see the details in the following:

“because of the low correlation of G3M-2 when compared with G3M-1 or G3M-3, the data for G3M-2 were discarded; Supplementary Fig. 12, Supplementary Table 13 and 14, Supplementary Methods”

Supplementary Fig. 12 Correlation heatmap of the G2M group and G3M group

23. Line 197: It is not clear if this is what is normally expected for the liver or something particular to the high lipid content of yellowstripe goby livers.

Response: Yes. High hepatic expression of genes associated with CHOL synthesis and transportation possibly promotes lipid droplets formation in the liver in yellowstripe goby.

24. Line 206: remove repeated “expression” and change high to highly.

Response: Thank you for your valuable comments, we accept the suggestions.

25. Line 208: remove “of”

Response: We fully agree with the reviewer. We have removed the word “of” from the manuscript.

26. Line 226: up-regulated compared to what?

Response: G3M was the control group. We have added some descriptions about the control group.

27. Line 232: earlier stated that this G3M data was discarded. Please reword the preceding statement to say it was not included in that analysis and state why.

Response: Thank you for your valuable comments. In fact, G3M-2 data was discarded, because of the low correlation of G3M-2 compared with G3M-1 and G3M-3 (Supplementary Fig. 12). We have added some descriptions about the data processing.

Please see the details in the following:

“because of the low correlation of G3M-2 when compared with G3M-1 or G3M-3, the data for G3M-2 were discarded; Supplementary Fig. 12, Supplementary Table 13 and 14, Supplementary Methods”

Supplementary Fig. 12 Correlation heatmap of the G2M group and G3M group

28. Line 239: in this in the yellowstripe goby?

Response: Yes. We have modified this sentence for clarity.

Please see the details in the following:

“Sequence analysis revealed that ABCA1 of yellowstripe goby has the largest copy number among known species”

29. Line 244: strange phrasing, what is meant by older evolutionary status.

Is this the ancestral copy? Which whole genome duplication is it likely derived from?

Response: Yes, ABCA1c is the ancestral copy. We have modified this sentence. We temporarily unable to confirm which time of whole genome duplication exactly derived *ABCA1c* based on the current information.

Please see the details in the following:

“Phylogenetic analysis revealed that ABCA1c was the ancestral copy (Fig. 4c)”

30. Line 253: all species sequenced in this study?

Response: Yes. We are sorry for our inaccurate statement in this sentence. We have modified this sentence.

Please see the details in the following:

“We found that yellowstripe goby possessed the largest number of TLR23 (15 copies) and tripartite motif containing (TRIM) family (234 members) in sequenced vertebrates so far”

31. Line 265: change benefit to be beneficial

Response: We fully agree with the reviewer. We have changed “benefit” to “be beneficial”.

32. Line 278: no rational given as to why might H2AFY be related to sex determination

Response: Thank you for your valuable comments. According to previous research, H2AFY might be participated in X-chromosome inactivation. We have added some discussion about function of H2AFY.

Please see the details in the following:

“We discovered that chromosome 5 of yellowstripe goby carries a 20.67-cM sex determination region that contains three genes that might be related to sex determination, namely, *MSL3*, *H2AFY* and *GALNT10-like*. *MSL3* contributes to overexpression of genes on the X chromosome of male *Drosophila*⁴⁴. Research on the structure of human *MSL3* has shown that it functions similar to the *Drosophila* *MSL3*⁴⁵, by binding to lysine 20 on the N-terminal tail of histone H4 to regulate the male-specific lethal complex on the X chromosome. H2AFY helps maintain X-chromosome inactivation⁴⁶. Thus, *MSL3* and H2AFY may be involved in yellowstripe goby sex determination.”

33. Line 286: A brief statement of the wider relevance of this study and any explanation as to why gobies would need such high lipids would be helpful here.

Response: Thank you for your valuable comments. We have added some discussion about this relevance.

Please see the details in the following:

“As a small bottom-layer fish inhabiting estuaries and intertidal zones, yellowstripe goby often faces sudden changes in salinity, temperature, water depth, light, and other factors. To facilitate rapid energy mobilization, yellowstripe goby has evolved a biological phenotype wherein the liver stores energy to cope with complex and changing living environments. Based on combined genome and transcriptome data, we identified three key metabolic pathways for lipid accumulation in the liver of yellowstripe goby, namely TAG synthesis, CHOL synthesis, and PC synthesis.”

34. Line 364: were the Iso-seq reads also from just liver?

Response: Thank you for your careful reading of our manuscript. We are

very sorry for our inaccurate statement in this paragraph. In fact, PacBio Iso-Seq full-length transcriptome data was not used in this study, we have removed the content from this manuscript.

35. Line 367-368: but were the transcripts really just from liver? This would mean many genes that are not expressed in the liver would be removed. Could this explain the low overlap in genes with other taxa?

Response: We are very sorry for our inaccurate statement in this paragraph. We just used the liver transcripts to evaluate the coverage of the reference genome, 99% sequences of liver transcripts were mappable. We have modified this paragraph.

Please see the details in the following:

“Furthermore, Illumina HiSeq RNA-Seq transcriptome data was used. HiSeq RNA-Seq reads from two liver tissues were mapped to the yellowstripe goby genome using Hisat2 (version 2.0.2). Finally, all the above data were combined to generate a comprehensive gene set using GLEAN.”

36. Line 379: please restate here which 3 species you refer to.

Response: Thank you for your valuable comments. These three species have been restated in manuscript.

Please see the details in the following:

“For the Ks analysis, we first used proteins sequences of the three species (*M.chulae*, *Boleophthalmus pectinirostris*, *Oncorhynchus mykiss*) to do blastp with themselves or between two species (Parameter: -m 8 -e 1e-5 -b 5 -v 5).”

37. Line 416: each of the 18 species?

Response: Yes, each of the 18 species were used to analysis the gene copy-number.

38. Line 438: less than or more than? What is a LD score - do you mean simply LD? How was LD calculated?

Response: We removed redundant markers with LD score>0.8. LD was expressed as r^2 , which was calculated by plink.

We tried our best to improve the manuscript and made some changes in the manuscript. These changes will not influence the content and framework of the paper. We appreciate for Editors/Reviewers’ warm work earnestly, and hope that the correction will meet with approval. Once again, thank you very much for your comments and suggestions.

Reviewers' comments:

Reviewer #1 (Remarks to the Author):

All of my questions have been addressed, and I recommend to publish this work.

Reviewer #2 (Remarks to the Author):

The authors have largely revised the manuscript in line with the reviewer comments. However, I find it strange that while so much of the manuscript is given over to the analysis of lipid gene expression there is no section covering this in the main text and that given in the supplementary appears incomplete. I am disappointed that this has not been addressed in the revisions and feel that this needs to be remedied. There is no comparison or discussion of the results in relation to those found for the recently published round or sand gobies despite the round goby study reporting similar results for the immune genes and both genome papers dealing with similar questions regarding adaptability of gobiidae. I think that the manuscript would benefit from this, but also some claims made in the text do not appear to be true in light of the results of these other genomic studies. This needs to be corrected before publication (see specific comments below).

There are some grammatical errors that have been introduced during revision, I have tried to detailed these below but I recommend the manuscript be carefully proofread before publication.

I have the following specific comments on the revised manuscript.

Line 20 "Differential expression of MGLL and CPT1..." different to what?

Line 22. "Expansion of the innate immune gene TLR23 and TRIM-family genes might provide insights for understanding the molecular mechanisms of sex determination and liver high-fat storage in marine fishes."

I don't understand the logic here. Why would this help understand mechanisms of sex determination? I feel that it will not be immediately clear how this is related to fat storage.

Line 26 "Compared with terrestrial vertebrates, sex determination mechanisms in fish have exhibited a high degree of plasticity and complexity" I don't think it is all terrestrial vertebrates that have simple sex determination mechanisms, complexity is also seen in amphibians and some reptiles. It is really birds and mammals for which this would be true.

Line 34 "there is little information" delete 'there is'

Line 35 "gobiids have has been discovered"

Line 36 "region" -> regions

Line 37 "fish is" -> fishes are

Line 38 "understanding its" -> understanding their

Line 38-40 The logic here is opaque to me. It is not clear why understanding the sex determining mechanisms will help to explain the global distribution. It may be hypothesized that sex determination transitions could be involved in speciation and could conceivably elevate species diversity as a result, but it is not yet clear if the gobiidae have experienced such transitions. I think this sentence is overstated and that there could be better ways to justify the study.

Line 47 showed -> showed

Line 48 lipid lipid - delete one

Line 48 Is this surprising when in the next sentence you reveal that this is already known for a model fish species and a major aquaculture species.

Line 54 – 57 It is not clear how these studies are similar to the one presented. This sentence is awkward and needs revision. The stickleback study is highly relevant to the idea that fat adaptation is required for transitions between marine and freshwater environments (which gobiidae have also done repeatedly). Sticklebacks have achieved this plasticity through copy number variation, which appears to be paralleled in the results presented here.

Line 58 delete as an energy storage

Line 73 I am not sure you mean “experimental evolution” and could leave simply as “evolution” or “adaptation”

Line 74 sequencing -> sequence

Line 76 the Sand goby genome is now also published: <https://doi.org/10.1111/jeb.13668>

Line 108 insert “generation”

Line 111 – comma misplaced -> 7,098

Line 132 rainbow trout also has a high repeat content (and likely other sequenced salmonids).

Line 144 “ultra-high-density genetic linkage map” with 9K markers this is quite moderate density in comparison to some other teleosts, for example the 56K map used to anchor the Rainbow trout genome.

Line 168 “However, the peak with yellow line continued into another peak with green line (Fig. 1d)” This is not at all clear, what were the yellow and green lines representing and what does it mean that they “run into each other” please make the text clear here especially since the figure legend is overly terse.

Line 173 “relationship” -> correspondence

Line 196, “We found 195 two male-determining genes and two female-determining genes,” only one female gene is described (FOXL2)

Line 200 “FOXL2 are critical genes for female determination and were all highly expressed” -> FOXL2 is a critical gene for female determination and was highly expressed

Line 206 “was observed for all seven genes (Supplementary Fig. 7, Supplementary methods).” Only 4 genes are discussed.

Line 211. I cannot find any analysis in the methods sections (main text or supplementary) that support the claim that high fat is dependent on genetic factors. There is no explanation of the rationale for this claim. The figure legends to the supplementary figures cited make no mention of what exactly in the figure suggest dependence on genetic factors.

Line 227 "because of the low correlation of G3M-2 when compared with G3M-1 or 226 G3M-3, the data for G3M-2 were discarded;"

It is unclear to me why would this low correlation occurred. Is this part of the natural variation, in which case it seems like discarding this data simply makes it more likely that a significant difference in expression between time points would be found. I would like to see an explanation of why this data set was so different and a better justification for its removal.

Line 256-257, 272 and 274. LD -> lipid droplet?

Line 279 compared to G3M

Line 282 FFAs -> Free fatty acids - easier to have full in the heading as not clear what this is without reading preceding section otherwise

Line 305 "We found that yellowstripe goby possessed the largest number of TLR23 (15 copies) and tripartite motif containing (TRIM) family members (234 members) in sequenced vertebrates so far"

But the round goby genome found 40 copies of TLR23. It would be helpful to add discussion about how these results compare to those reported for the round goby. Larger copy numbers are found in other fish species see those referred to in the round goby genome paper and "Solbakken MH, Voje KL, Jakobsen KS, Jentoft S. Linking species habitat and past palaeoclimatic events to evolution of the teleost innate immune system. Proc R Soc B Biol Sci. 2017;284:20162810."

Line 328 "The average marker distance was 0.32 cM, which is higher than that for most fish species at present"

Higher -> lower

Is this true for aquaculture species with high density maps like Atlantic salmon or rainbow trout? Maybe a qualifier such as "for a non-model and non-aquaculture species" is needed here?

Line 349 be participating the formation -> be participating in the formation

Line 358 "Thus, DMRT1 and GDSF might be master sex-determining genes in yellowstripe goby." Do they located into the sex specific region on chr 5?

Line 412 "Genome analysis revealed that the innate immunity gene TLR23 is expanded in yellowstripe goby, with 15 copies, which is the highest copy number among any species reported to date."

See earlier comment, this appears not to be true with many teleost having higher copy numbers, notably the round goby.

Line 441. I think it is strange given how much of manuscript describes the fat results that there is not section for this in the main methods for the paper. I realise there is information in the supplementary material but since the manuscript relies heavily on this work I would expect it to be in the main text.

Line 580 "with a linkage disequilibrium score of" I think this should just be linkage disequilibrium to avoid confusion.

Line 592. The phenotype data would also need to be released in order to be able to repeat the analysis for detecting the sex region.

Supplementary Line 224 -227. The main text refers to these 2 and 3 month samples as separate replicates, but it is not clear how this data were obtained if the samples were pooled prior to sequencing. No information on how these samples were analysed is provided making it even harder to work out what actually was done here.

Supplementary Line 232 Although the analysis methods have been described in a previous publication I would expect a brief summary to be provided here, especially as this is supplementary methods and so space is not limiting.

Responses to the reviewers

We would like to thank the Reviewers again for your thoughtful review and comments on the manuscript, entitled “**Whole-genome sequencing reveals sex determination and liver high-fat storage mechanisms of yellowstripe goby (*Mugilogobius chulae*)**”. Those comments are all valuable and very helpful for revising and improving our paper. We have studied comments carefully and have made correction which we hope meet with approval. **Revised portion are marked in red in the paper.** The point to point responds to the reviewer’s comments are listed as following:

Reviewers' comments:

Reviewer #1 (Remarks to the Author):

All of my questions have been addressed, and I recommend to publish this work.

Reviewer #2 (Remarks to the Author):

The authors have largely revised the manuscript in line with the reviewer comments. However, I find it strange that while so much of the manuscript is given over to the analysis of lipid gene expression there is no section covering this in the main text and that given in the supplementary appears incomplete. I am disappointed that this has not

been addressed in the revisions and feel that this needs to be remedied. There is no comparison or discussion of the results in relation to those found for the recently published round or sand gobies despite the round goby study reporting similar results for the immune genes and both genome papers dealing with similar questions regarding adaptability of gobiidae. I think that the manuscript would benefit from this, but also some claims made in the text do not appear to be true in light of the results of these other genomic studies. This needs to be corrected before publication (see specific comments below).

There are some grammatical errors that have been introduced during revision, I have tried to detailed these below but I recommend the manuscript be carefully proofread before publication.

I have the following specific comments on the revised manuscript.

Question 1: Line 20 “Differential expression of MGLL and CPT1...” different to what?

Response: Thank you for your patient review and valuable suggestions.

We have modified this paragraph for better understanding.

Please see the details in the following:

“The changes in the expression patterns of *MGLL* and *CPT1* at different development stage of the liver, ...”

Question 2: Line 22. “Expansion of the innate immune gene TLR23 and TRIM-family genes might provide insights for understanding the molecular mechanisms of sex determination and liver high-fat storage in marine fishes.”

I don't understand the logic here. Why would this help understand mechanisms of sex determination? I feel that it will not be immediately clear how this is related to fat storage.

Response: We have modified this paragraph for better understanding. Please see the details in the following:

“Abstract:, The changes in the expression patterns of *MGLL* and *CPT1* at different development stage of the liver, and the expansion of the ABCA1 gene, innate immune gene *TLR23*, and *TRIM* family genes may help in balancing high-fat storage in hepatocytes and steatohepatitis. These results may provide insights into understanding the molecular mechanisms of sex determination and high-fat storage in the liver of marine fishes.”

Question 3: Line 26 “Compared with terrestrial vertebrates, sex determination mechanisms in fish have exhibited a high degree of plasticity and complexity” I don't think it is all terrestrial vertebrates that have simple sex determination mechanisms, complexity is also seen in amphibians and some reptiles. It is really birds and mammals for which

this would be true.

Response: As suggested by the reviewer, we have modified this sentence for more readable.

Please see the details in the following:

“Compared with **most mammals.....**”

Question 4

(1) Line 34 “there is little information” delete ‘there is’

(2) Line 35 “gobiids have has been discovered”

(3) **Question 6** Line 36 “region” -> regions

(4) **Question 7** Line 37 “fish is” -> fishes are

(5) **Question 8** Line 38 “understanding its” -> understanding their

Response: We fully agree with the reviewer’s suggestion. As suggested by reviewer, we have revised this paragraph.

Please see the details in the following:

“Especially in gobiid fish, one of the largest fish families, comprising more than 2,000 species, little information about the sex determination mechanism **has** been discovered, **and** no sex determination genes or sex determination **regions** have been identified to date.”

“understanding **their** sex determination mechanism is of.....”

Question 5: Line 38-40 The logic here is opaque to me. It is not clear

why understanding the sex determining mechanisms will help to explain the global distribution. It may be hypothesized that sex determination transitions could be involved in speciation and could conceivably elevate species diversity as a result, but it is not yet clear if the gobiidae have experienced such transitions. I think this sentence is overstated and that there could be better ways to justify the study.

Response: We are very sorry for our inaccurate statement in this paragraph. We have deleted “global distribution” from this sentence.

Please see the details in the following:

“understanding **their** sex determination mechanism is of great significance **in** revealing the adaptive strategies and providing invaluable insights on the **evolution of** sex determination in teleosts.”

Question 6:

(1) Line 47 showed -> showed

(2) Line 48 lipid lipid - delete one

Response: Thank you for your valuable comments. We have revised this sentence.

Please see the details in the following:

“Most fishes, such as zebrafish (*Danio rerio*)¹⁴ and cavefish (*Astyanax mexicanus*)¹⁶, **show** deposition of lipids [mainly triglycerides (TAG)] in mesentery and viscera.”

Question 7:

Line 48 Is this surprising when in the next sentence you reveal that this is already known for a model fish species and a major aquaculture species.

Response: Thank you for your valuable comments. We have modified this sentence as following: “**However**, in majority of the gobiids,

Question 8:

Line 54 – 57 It is not clear how these studies are similar to the one presented. This sentence is awkward and needs revision. The stickleback study is highly relevant to the idea that fat adaptation is required for transitions between marine and freshwater environments (which gobiidae have also done repeatedly). Sticklebacks have achieved this plasticity through copy number variation, which appears to be paralleled in the results presented here.

The stickleback study is highly relevant to the idea that fat adaptation is required for transitions between marine and freshwater environments (which gobiidae have also done repeatedly). Sticklebacks have achieved this plasticity through copy number variation, which appears to be paralleled in the results presented here

Response: Thank you for your valuable and thoughtful comments.

We have revised the section as following:

“stickleback lineages (*Gasterosteus aculeatus* species complex), which have evolved different copy numbers of lipid metabolism-related genes, such as docosahexaenoic acid biosynthesis-related genes, to achieve transitions between marine and freshwater environments.”

Question 9: Line 58 delete as an energy storage

Response: Thank you for your valuable comments, we have deleted this phrase from the manuscript.

Question 10: Line 73 I am not sure you mean “experimental evolution” and could leave simply as “evolution” or “adaptation”

Response: We have replaced “experimental evolution adaptation” by “adaptation”.

Question 11: Line 74 sequencing -> sequence

Response: We fully agree with the reviewer’s suggestion. We have changed “sequencing” to “sequence”.

Question 12: Line 76 the Sand goby genome is now also published:

<https://doi.org/10.1111/jeb.13668>

Response: Thanks for your reminder. We have quoted this article in the manuscript.

“Only a few gobiids genomes have been sequenced, such as of round goby (*Neogobius melanostomus*)⁸, mudskippers (*Boleophthalmus pectinirostris*, *Periophthalmodon schlosseri*, *Periophthalmus magnuspinnatus*, *Scartelaos histophorus*)¹⁰ and sand goby (*Pomatoschistus minutus*)²⁹”.

Reference:

[29] Leder, E. H., André, C., Le Moan, A., Töpel, M., Blomberg, A., Havenhand, J. N., ... & Svensson, O. (2020). Post-glacial establishment of locally adapted fish populations over a steep salinity gradient. *Journal of Evolutionary Biology*. 00:1–19.

Question 13:

Line 108 insert “generation”

Line 111 comma misplaced -> 7,098

Response: We have inserted the word and modified this mistake in the manuscript.

Question 14: Line 132 rainbow trout also has a high repeat content (and likely other sequenced salmonids).

Response: Thank you for your valuable suggestions. Actually, the rainbow trout genome had an overall repeat content of 37.8% [1], which is lower to that in yellowstripe goby, therefore, we have kept the original statement.

Reference:

[1] Berthelot, C. et al. The rainbow trout genome provides novel insights into evolution after whole-genome duplication in vertebrates. *Nature Communications* 5(2014).

Question 15: Line 144 “ultra-high-density genetic linkage map” with 9K markers this is quite moderate density in comparison to some other teleosts, for example the 56K map used to anchor the Rainbow trout genome.

Response: We have removed “ultra” in the manuscript.

Question 16: Line 168 “However, the peak with yellow line continued into another peak with green line (Fig. 1d)” This is not at all clear, what were the yellow and green lines representing and what does it mean that they “run into each other” please make the text clear here especially since the figure legend is overly terse.

Response: We are very sorry for our inaccurate statement in this paragraph. As suggested by the reviewer, we have revised this paragraph and the figure legend of Fig 1.

Please see the details in the following:

“The spectrum of synonymous substitutions (Ks) among yellowstripe goby, mudskipper, and *Oncorhynchus mykiss* showed peaks at 0.5 for gobiids (yellowstripe goby versus mudskipper, yellow curve in Fig. 1d), which were close to those for trout (*O. mykiss* versus *O. mykiss*,

pink curve in Fig. 1d), that has undergone four whole-genome duplication events (WGD) (Fig. 1d). However, the peak with yellow line continued into another peak with green line (yellowstripe goby versus yellowstripe goby) (Fig. 1d).”

“Fig.1 d The third genome duplications in the yellowstripe goby genome was identified by Ks analyses; the pink curve represents the fourth genome duplication event of *O. mykiss*, green and orange curve represents the third genome duplication event of yellowstripe goby and mudskipper, yellow and blue curve represents the interspecific differentiation.”

Question 17: Line 173 “relationship” -> correspondence

Response: As suggested by the reviewer, we have replaced “relationship” by “correspondence”.

Question 18: Line 196, “We found two male-determining genes and two female-determining genes,” only one female gene is described (FOXL2)

Response: We are very sorry for our inaccurate statement in this paragraph. We have changed “two female-determining genes” to “one female-determining gene”.

Question 19: Line 200 “FOXL2 are critical genes for female

determination and were all highly expressed” -> FOXL2 is a critical gene for female determination and was highly expressed

Response: As suggested by the reviewer, we have revised the grammar.

Question 20: Line 206 “was observed for all seven genes (Supplementary Fig. 7, Supplementary methods).” Only 4 genes are discussed.

Response: Actually we had discussed six genes in the manuscript (MSL3, H2AFY, GLANT10-like, DMRT1, GSDF and FOXL2, respectively). The remaining one, FOXL3, may be related to testis and ovary development, had not been discussed. The relationship between FOXL3 and sex determination in yellowstripe goby might be worthy of further investigation.

Question 21: Line 211. I cannot find any analysis in the methods sections (main text or supplementary) that support the claim that high fat is dependent on genetic factors. There is no explanation of the rationale for this claim. The figure legends to the supplementary figures cited make no mention of what exactly in the figure suggest dependence on genetic factors.

Response: We appreciate the reviewer for this important comment. Actually, high fat deposition may be a normal physiological phenomenon

in yellowstripe goby which were supported by the histological observation, fasting experiment and feeding experiments at different nutrient levels. We have replaced “genetic factors” by “a normal physiological phenomenon”.

Please see the details in the following:

“Histological observation of the liver of yellowstripe goby showed that high-fat deposition in the liver might be a normal physiological phenomenon,

Question 22: Line 227 “because of the low correlation of G3M-2 when compared with G3M-1 or G3M-3, the data for G3M-2 were discarded;”

It is unclear to me why would this low correlation occurred. Is this part of the natural variation, in which case it seems like discarding this data simply makes it more likely that a significant difference in expression between time points would be found. I would like to see an explanation of why this data set was so different and a better justification for its removal.

Response: We are sorry for the insufficient statement about the sample preparation. Usually, in order to avoid errors, three duplicates in each group were used in transcriptome sequencing. In our study, G3M-1, G3M-2 and G3M-3 were the replicate samples of G3M group. The low correlation of G3M-2 when compared with G3M-1 or G3M-3 might be related with the process of RNA extraction, cDNA library preparation or

sequencing. Hence, the sequencing data from G3M-1 and G3M-3 should be more representative of the reality of G3M group. We have revised the section of “Transcriptome sequencing” in Supplementary methods.

Please see the details in the following:

“The samples used for liver fat-accumulation transcriptome analysis were from a closed colony of 2-month-old and 3-month-old yellowstripe goby, with 18 and 9 individuals, respectively. The fish were bred in our laboratory. The livers of three 2-month-old and three 3-month-old fishes were pooled into two samples each, to give a total of six samples (three duplicates in each group). The 2-month-old sample was labeled G2M (duplicate samples: G2M-1, G2M-2 and G2M-3), and the 3-month-old sample was labeled G3M (duplicate samples: G3M-1, G3M-2 and G3M-3).”

Question 23: Line 256-257, 272 and 274. LD -> lipid droplet?

Response: As suggested by the reviewer, we have replaced LD by lipid droplet.

Question 24: Line 279 compared to G3M

Response: As suggested by the reviewer, we have revised this sentence.

Question 25: Line 282 FFAs -> Free fatty acids - easier to have full in

the heading as not clear what this is without reading preceding section otherwise

Response: As suggested by the reviewer, we have replaced “FFAs” by “Free fatty acid”.

Question 26: Line 305 “We found that yellowstripe goby possessed the largest number of TLR23 (15 copies) and tripartite motif containing (TRIM) family members (234 members) in sequenced vertebrates so far”

But the round goby genome found 40 copies of TLR23. It would be helpful to add discussion about how these results compare to those reported for the round goby Larger copy numbers are found in other fish species see those referred to in the round goby genome paper and “Solbakken MH, Voje KL, Jakobsen KS, Jentoft S. Linking species habitat and past palaeoclimatic events to evolution of the teleost innate immune system. Proc R Soc B Biol Sci. 2017;284:20162810.”

Response: We appreciate the reviewer for this important comment. As suggested by the reviewer, we have revised this sentence and added more contents in supplementary Table 16.

Please see the details in the following:

“We discover an expansion of TLR23 (15 copies) and tripartite motif containing (TRIM) family members (234 members) in yellowstripe goby (Supplementary Table 16)”.

Supplementary Table 16 The copy number of TLR23 gene in different species^[1-3]

Species	TLR23
Yellowstripe goby (M. chulae)	15
Round goby (Neogobius melanostomus)	40
Bluespotted mudskipper (Boleophthalmus pectinirostris)	12
Minute mudskipper (Periophthalmus magnuspinnatus)	7
Giant mudskipper (Periophthalmodon schlosseri)	9
Walking goby (Scartelaos histophorus)	6
European perch (Perca fluviatilis)	17
Kissing Gourami (Helostoma temminckii)	14
Glacier lanternfish (Benthoosema glaciale)	49
Mexican cave tetra (Astyanax mexicanus)	3
Amazon molly (Poecilia formosa)	3
Southern platyfish (Xiphophorus maculatus)	1
Nile tilapia (Oreochromis niloticus)	1
Japanese pufferfish (Fugu rubripes)	1
Freshwater pufferfish (Tetraodon nigroviridis)	1
Zebrafish (Danio rerio)	0
Medaka (Oryzias latipes)	0
coelacanth (Latmeria chalumnae)	0
Mouse (Mus musculus)	0
Human (Homo sapiens)	0

We have also enriched the discussion. Please see the details in the following:

“Genome analysis revealed that the innate immunity gene *TLR23* is expanded in yellowstripe goby, with 15 copies, which is higher than that in most other fish, except round goby (40 copies)⁸, Glacier lanternfish (*Benthoosema glaciale*) (49 copies)⁸ and European perch (*Perca fluviatilis*) (17 copies)⁸. In teleosts, *TLR* expansions might correlate with the survival and successful radiation of this lineage⁶⁴. The expansion of *TLR23* in yellowstripe goby may help it to adapt to the complex environment, for instance, by reducing the amplification effect of the external environment on steatohepatitis.”

Reference:

- [1] Adrian-Kalchauer, I., Blomberg, A., Larsson, T., Musilova, Z., Peart, C. R., Pippel, M., ... & Rosenblad, M. A. (2020). The round goby genome provides insights into mechanisms that may facilitate biological invasions. *BMC biology*, 18(1), 1-33.
- [2] Solbakken, M. H., Voje, K. L., Jakobsen, K. S., & Jentoft, S. (2017). Linking species habitat and past palaeoclimatic events to evolution of the teleost innate immune system. *Proceedings of the Royal Society B: Biological Sciences*, 284(1853), 20162810.
- [3] Solbakken, M. H., Tørresen, O. K., Nederbragt, A. J., Seppola, M., Gregers, T. F., Jakobsen, K. S., & Jentoft, S. (2016). Evolutionary redesign of the Atlantic cod (*Gadus morhua* L.) Toll-like receptor repertoire by gene losses and expansions. *Scientific reports*, 6(1), 1-14.

Question 27: Line 328 “The average marker distance was 0.32 cM, which is higher than that for most fish species at present”

Higher -> lower

Is this true for aquaculture species with high density maps like Atlantic salmon or rainbow trout? Maybe a qualifier such as “for a non-model and non-aquaculture species” is needed here?

Response: Thank you for your valuable comments. We have revised this sentence.

Please see the details in the following:

“The average marker distance was 0.32 cM, which is **lower** than that for most **non-model and non-aquaculture species**, including *Cyprinus carpio haematopterus* (0.57 cM)⁴¹, *Nibeal albiflora* (0.47 cM)⁴², *Larmichthys crocea* (0.36 cM)⁴³, and *Pseudobagrus ussuriensis* (0.36 cM)⁴⁴.”

Question 28: Line 349 be participating the formation -> be participating

in the formation

Response: As suggested by reviewer, we have modified this paragraph.

Question 29: Line 358 “Thus, DMRT1 and GDSF might be master sex-determining genes in yellowstripe goby.” Do they located into the sex specific region on chr 5?

Response: DMRT was located at Chr 3 and GSDF could not be located temporarily. Because DMRT1 and GSDF were identified by RNA-seq, these genes might not be located at Chr 5.

Question 30: Line 412 “Genome analysis revealed that the innate immunity gene TLR23 is expanded in yellowstripe goby, with 15 copies, which is the highest copy number among any species reported to date.”

See earlier comment, this appears not to be true with many teleost having higher copy numbers, notably the round goby.

Response: Thank you for your valuable comments. Please refer to Question 26 for a detailed reply.

Question 31: Line 441. I think it is strange given how much of manuscript describes the fat results that there is not section for this in the main methods for the paper. I realise there is information in the supplementary material but since the manuscript relies heavily on this

work I would expect it to be in the main text.

Response: Thank you for your valuable comments. We have moved “Liver tissue sections of yellowstripe goby”, “Determination of the hepatic lipid composition” and “Validation of differentially expressed genes by quantitative real-time PCR” from Supplementary Material section to main Methods.

Question 32: Line 580 “with a linkage disequilibrium score of” I think this should just be linkage disequilibrium to avoid confusion.

Response: As suggested by reviewer, we have changed “with a linkage disequilibrium score of” to “linkage disequilibrium”.

Question 33: Line 592. The phenotype data would also need to be released in order to be able to repeat the analysis for detecting the sex region.

Response: Thank you for your valuable suggestion. We have added individual ID which used for identifying sex determination region in Supplementary Table 19. All raw data of these individuals have been deposited in NCBI under project PRJNA642226. Anyone could repeat the analysis by these data.

Please see the details in the following:

Supplementary Table 19 Individual ID for identifying sex determination region

Serial number	Parent ID			
	Female parent ID		Male parent ID	
1	WHYD17055725		WHYD17055726	
Serial number	Offspring ID			
	Female		Male	
1	WH1705002044	WHYD17055614	WHYD17055511	WHYD17055593
2	WHYD17055513	WHYD17055615	WHYD17055512	WHYD17055594
3	WHYD17055514	WHYD17055619	WH1705002047	WHYD17055595
4	WHYD17055516	WHYD17055620	WH1705002049	WHYD17055596
5	WHYD17055519	WHYD17055621	WH1705002050	WHYD17055597
6	WH1705002059	WHYD17055625	WH1705002051	WHYD17055605
7	WHYD17055521	WHYD17055627	WH1705002053	WHYD17055612
8	WHYD17055523	WHYD17055628	WHYD17055518	WHYD17055616
9	WHYD17055528	WHYD17055636	WHYD17055522	WHYD17055618
10	WHYD17055529	WHYD17055639	WHYD17055524	WHYD17055622
11	WHYD17055534	WHYD17055640	WHYD17055530	WHYD17055624
12	WHYD17055536	WHYD17055641	WHYD17055531	WHYD17055626
13	WHYD17055540	WHYD17055645	WHYD17055532	WHYD17055629
14	WHYD17055542	WHYD17055650	WHYD17055533	WHYD17055630
15	WH1705002089	WHYD17055661	WHYD17055536	WHYD17055637
16	WHYD17055548	WHYD17055667	WHYD17055538	WHYD17055642
17	WH1705002093	WHYD17055669	WHYD17055539	WHYD17055649
18	WH1705002094	WHYD17055673	WHYD17055543	WHYD17055657
19	WHYD17055551	WHYD17055676	WHYD17055544	WHYD17055662
20	WHYD17055552	WHYD17055680	WHYD17055545	WHYD17055665
21	WHYD17055553	WHYD17055685	WHYD17055547	WHYD17055668
22	WHYD17055555	WHYD17055688	WH1705002092	WHYD17055677
23	WHYD17055557	WHYD17055690	WHYD17055549	WHYD17055684
24	WHYD17055560	WHYD17055691	WHYD17055550	WHYD17055689
25	WHYD17055568	WH1705002240	WHYD17055554	WHYD17055699
26	WHYD17055571	WHYD17055693	WHYD17055558	WHYD17055707
27	WHYD17055574	WHYD17055695	WHYD17055559	WHYD17055709
28	WHYD17055577	WHYD17055698	WHYD17055561	WHYD17055714
29	WHYD17055583	WHYD17055702	WHYD17055562	WHYD17055717
30	WHYD17055586	WHYD17055703	WHYD17055566	WHYD17055718
31	WHYD17055592	WHYD17055705	WHYD17055570	WHYD17055722
32	WHYD17055600	WHYD17055710	WHYD17055580	/
33	WHYD17055601	WHYD17055711	WHYD17055584	/
34	WHYD17055603	WHYD17055715	WHYD17055587	/
35	WHYD17055610	WHYD17055716	WHYD17055589	/
36	WHYD17055613	WHYD17055721	WHYD17055591	/

Question 34: Supplementary Line 224 -227. The main text refers to these 2 and 3 month samples as separate replicates, but it is not clear how this data were obtained if the samples were pooled prior to sequencing. No information on how these samples were analysed is provided making it even harder to work out what actually was done here.

Response: We are very sorry for our inaccurate statement in this paragraph. We have added some details in this paragraph. Please refer to Question 22 for a detailed reply.

Please see the details in the following:

“The samples used for liver fat-accumulation transcriptome analysis were from a closed colony of 2-month-old and 3-month-old yellowstripe goby, with 18 and 9 individuals, respectively. The fish were bred in our laboratory. The livers of three 2-month-old and three 3-month-old fishes were pooled into two samples each, to give a total of six samples (three duplicates in each group). The 2-month-old sample was labeled G2M (duplicate samples: G2M-1, G2M-2 and G2M-3), and the 3-month-old sample was labeled G3M (duplicate samples: G3M-1, G3M-2 and G3M-3).”

Question 35: Supplementary Line 232 Although the analysis methods have been described in a previous publication I would expect a brief summary to be provided here, especially as this is supplementary methods and so space is not limiting.

Response: As suggested by reviewer, we have added some details this paragraph.

Please see the details in the following:

“Raw reads were processed by removing the adapters along with low-quality sequences. All unigenes were aligned to the Nr, Nt, Swissprot, Kyoto Encyclopedia of Genes and Genomes (KEGG), and Gene Ontology (GO) databases with the Basic Local Alignment Search Tool program. A robust and efficient stack memory management method was used to count the reads mapped to each unigene. The reads per kilobase per million mapped reads (RPKM) value was measured based on transcript length and read counts mapped to the transcript. Differentially expressed genes were detected with the DEGSeq R package.”

REVIEWERS' COMMENTS:

Reviewer #2 (Remarks to the Author):

The authors have addressed most of my concerns with the manuscript and I thank them for their work. I just have a couple of minor points remaining. I am sure that the editor can deal with these small points.

"Question 14: Line 132 rainbow trout also has a high repeat content (and likely other sequenced salmonids).

Response: Thank you for your valuable suggestions. Actually, the rainbow trout genome had an overall repeat content of 37.8% [1], which is lower to that in yellowstripe goby, therefore, we have kept the original statement.

Reference:

[1] Berthelot, C. et al. The rainbow trout genome provides novel insights into evolution after whole-genome duplication in vertebrates. *Nature Communications* 5(2014)."

The Berthelot et al genome was an early attempt to sequence this complex genome, a more complete rainbow trout genome was released on NCBI in 2017 and published in 2019 (<https://doi.org/10.1038/s41559-019-1044-6>). This version has an estimated 57.1% repeat content, so very similar to the estimate from Atlantic salmon (58%). It is not essential to include reference to this, but all salmonids except grayling likely share very high repeat contents. This because they share two repeat expansion that occurred following the salmonid specific whole genome duplication event. However, more importantly other unrelated fish species are also known to have high repeat contents, such as the killifish *Nothobranchius furzeri*, which has an estimated repeat content of 64.6% (<https://doi.org/10.1016/j.cell.2015.10.071>). Given which, the repeat content in the goby does not appear particularly remarkable for a teleost (though this likely varies very widely). The current wording suggest few teleosts have high repeat contents, I suspect this is not the case.

"Question 29: Line 358 "Thus, DMRT1 and GDSF might be master sex-determining genes in yellowstripe goby." Do they located into the sex specific region on chr 5?

Response: DMRT was located at Chr 3 and GDSF could not be located temporarily. Because DMRT1 and GDSF were identified by RNA-seq, these genes might not be located at Chr 5."

If the master regulator is expected to be in the sex-specific region, this would seem to reject DMRT1 being the master regulator as suggested in the text, though undoubtedly it plays a role in the sex determination cascade. If GDSF is regulated by DMRT1 it is unclear how it would be the "master regulator". As many genes involved in the sex determination cascade can be used as the master regulator, I am not sure that finding them expressed in either testis or ovaries (FOXL2) is particularly informative about whether they are master regulators. It matters which gene initiates the process, which can't be established from the presented RNAseq data. Perhaps you could be more cautious about suggesting these genes as master regulators since there doesn't seem to be any evidence to suggest that they are in the presented work.

Responses to the reviewers

We would like to thank the Reviewer #2 again for your thoughtful review and comments on the manuscript, entitled “**Whole-genome sequencing reveals sex determination and liver high-fat storage mechanisms of yellowstripe goby (*Mugilogobius chulae*)**”. We have studied comments carefully and have made correction which we hope meet with approval. **Revised portion are marked in red in the paper.** The point to point responds to the reviewer’s comments are listed as following:

Reviewers' comments:

Reviewer #2 (Remarks to the Author):

The authors have addressed most of my concerns with the manuscript and I thank them for their work. I just have a couple of minor points remaining. I am sure that the editor can deal with these small points.

Question 1:

“Question 14: Line 132 rainbow trout also has a high repeat content (and likely other sequenced salmonids).

Response: Thank you for your valuable suggestions. Actually, the rainbow trout genome had an overall repeat content of 37.8% [1], which is lower to that in yellowstripe goby, therefore, we have kept the original

statement.

Reference:

[1] Berthelot, C. et al. The rainbow trout genome provides novel insights into evolution after whole-genome duplication in vertebrates. *Nature Communications* 5(2014).”

The Berthelot et al genome was an early attempt to sequence this complex genome, a more complete rainbow trout genome was released on NCBI in 2017 and published in 2019 (<https://doi.org/10.1038/s41559-019-1044-6>). This version has an estimated 57.1% repeat content, so very similar to the estimate from Atlantic salmon (58%). It is not essential to include reference to this, but all salmonids except grayling likely share very high repeat contents. This because they share two repeat expansion that occurred following the salmonid specific whole genome duplication event. However, more importantly other unrelated fish species are also known to have high repeat contents, such as the killifish *Nothobranchius furzeri*, which has an estimated repeat content of 64.6% (<https://doi.org/10.1016/j.cell.2015.10.071>). Given which, the repeat content in the goby does not appear particularly remarkable for a teleost (though this likely varies very widely). The current wording suggest few teleosts have high repeat contents, I suspect this is not the case.

Response: We are very sorry for our inaccurate statement in this

paragraph. As suggested by reviewer, we have revised this paragraph.

Please see the details in the following:

“The yellowstripe goby genome had an overall repeat content of 42.56%, which is similar to that in mudskipper¹⁰ and round goby, but **lower than that in** zebrafish³⁰ (52.2%) and Atlantic salmon³⁹ (58%).”

Question 2:

“Question 29: Line 358 “Thus, DMRT1 and GDSF might be master sex-determining genes in yellowstripe goby.” Do they located into the sex specific region on chr 5?

Response: DMRT was located at Chr 3 and GDSF could not be located temporarily. Because DMRT1 and GDSF were identified by RNA-seq, these genes might not be located at Chr 5.”

If the master regulator is expected to be in the sex-specific region, this would seem to reject DMRT1 being the master regulator as suggested in the text, though undoubtedly it plays a role in the sex determination cascade. If GDSF is regulated by DMRT1 it is unclear how it would be the “master regulator”. As many genes involved in the sex determination cascade can be used as the master regulator, I am not sure that finding them expressed in either testis or ovaries (FOXL2) is particularly informative about whether they are master regulators. It matters which gene initiates the process, which can't be established from the presented

RNAseq data. Perhaps you could be more cautious about suggesting these genes as master regulators since there doesn't seem to be any evidence to suggest that they are in the presented work.

Response: Thank you for your valuable suggestion. We have revised this paragraph. Please see the details in the following:

“Thus, *DMRT1* and *GSDF* might be **potential** sex-determining genes in yellowstripe goby. In addition, we found that **the *FOXL2* was** highly expressed specifically in the yellowstripe goby ovary. *FOXL2* is a key gene in ovarian differentiation and development and may **participate in** female sex determination in yellowstripe goby.”